# Bridging the Gap Between Preference Alignment and Machine Unlearning

## Abstract

Despite advances in Preference Alignment (PA) for Large Language Models (LLMs), mainstream methods like Reinforcement Learning with Human Feedback (RLHF) face notable challenges. These approaches require high-quality datasets of positive preference examples, which are costly to obtain and computationally intensive due to training instability, limiting their use in low-resource scenarios. The LLM unlearning technique presents a promising alternative by directly removing the influence of negative examples. However, current research has primarily focused on empirical validation, lacking systematic quantitative analysis. To bridge this gap, we propose a framework to explore the relationship between PA and LLM unlearning. Specifically, we introduce a bi-level optimization-based method to quantify the impact of unlearning specific negative examples on PA performance. Our analysis reveals that not all negative examples contribute equally to alignment improvement when unlearned, and the effect varies significantly across examples. Building on this insight, we pose a crucial question: how can we optimally select and weight negative examples for unlearning to maximize PA performance? To answer this, we propose a framework called Unlearning to Align (U2A), which leverages bi-level optimization to efficiently select and unlearn examples for optimal PA performance. We validate the proposed method through extensive experiments, with results confirming its effectiveness. Our code is available at https://anonymous.4open.science/r/U2A-9E75.

## 1 Introduction

Despite the strong performance of Large Language Models (LLMs) in predicting the next token, their generated content often exhibits biases, factual inaccuracies, and other undesirable behaviors (Bai et al., 2022; Casper et al., 2023). Preference Alignment (PA) has been proposed to address these issues by guiding LLMs to generate responses aligned with human preferences, such as fairness and helpfulness (Ziegler et al., 2019; Stiennon et al., 2020). This approach uses datasets of human-annotated preferred and non-preferred responses to optimize the model. Reinforcement Learning from Human Feedback (RLHF) is the primary method for achieving PA (Wu et al., 2024; Azar et al., 2024), involving the training of a reward model on human preference data and optimizing the LLM using algorithms like Proximal Policy Optimization (PPO) (Schulman et al., 2017) or Direct Preference Optimization (DPO) (Rafailov et al., 2024). While RLHF shows strong performance across diverse applications, such as programming and creative writing, *it relies on costly large-scale preference-aligned datasets, especially for positive examples* (Yao et al., 2024). Additionally, RLHF training is computationally intensive and prone to instability (Liu et al., 2024c; Zhou et al., 2024b), posing challenges for low-resource alignment scenarios.

As a key technique aimed at protecting user privacy, Machine Unlearning (MU) in LLMs offers a novel solution to the aforementioned challenges (Liu et al., 2024d; Yao et al., 2024). This technique enables the removal of specific user data from pre-trained LLMs without requiring a complete retraining. By facilitating the unlearning of negative examples, this promotes PA while addressing the high costs and difficulties associated with acquiring positive examples for standard RLHF. Unlike RLHF, *LLM unlearning requires only negative examples, which are typically easier and cheaper to collect via mechanisms like user reports or red team testing*. For unaligned pre-trained models, identifying counterexamples can be highly automated, further reducing data collection costs. Additionally, the computational overhead of unlearning is comparable to fine-tuning and significantly

lower than RLHF's full training process, making it a practical approach for achieving alignment in low-resource scenarios.

Existing studies (Feng et al., 2024; Liu et al., 2024d; Yao et al., 2024) have validated the effectiveness of achieving model alignment through the unlearning of negative examples, highlighting the potential of integrating MU with PA. However, these studies primarily rely on experimental demonstrations, lacking in-depth quantitative analysis. For instance, the quantitative impact of unlearning specific samples on PA remains unclear. Additionally, critical questions such as which examples should be unlearned to maximize alignment and how to optimally select subsets of examples for unlearning to achieve the best outcomes remain unresolved. These gaps underscore a theoretical and practical disconnect between MU and PA. Addressing these challenges requires the development of a comprehensive analytical framework to unify these two domains and facilitate a deeper understanding of their intrinsic connections.

To address the identified challenges, we first develop a special bi-level optimization framework to quantify how unlearning specific negative samples impacts model PA performance. In particular, the inner optimization focuses on unlearning the target sample, while the outer optimization assesses the resulting change in PA performance. After further analysis, we find that not all negative examples contribute to PA improvement, with the degree of impact varying across examples. Meanwhile, the impact is influenced by the unlearning weights. This suggests that indiscriminately applying unlearning to all negative examples fails to achieve optimal PA performance. To address this, we propose a framework called Unlearning to Align (U2A), based on bi-level optimization, to strategically select samples and determine optimal unlearning weights. Further convergence and computational complexity analysis indicate that our proposed method demonstrates good applicability and efficiency in LLMs. This framework bridges the gap between MU and PA, offering a systematic approach to their integration. We summarize the main contributions of this paper as follows:

- We propose a special bi-level optimization framework to measure the impact of unlearning specific samples on PA performance, bridging the gap between MU and PA.
- We find that unlearning all negative examples does not always benefit PA, as their contributions to PA improvement vary and can be adjusted through unlearning weights.
- We propose the U2A framework, leveraging bi-level optimization to select and weight negative examples for unlearning, thereby maximizing PA performance.
- We conduct extensive evaluations on multiple models and real-world datasets, and the experimental results demonstrate the effectiveness of our method.

## 2 PRELIMINARY

Given a training set $\mathcal{D}_t = \{\boldsymbol{x}^1, \boldsymbol{x}^2, \ldots, \boldsymbol{x}^{N_t}\}$, where $\boldsymbol{x}^i = \{x_1, x_2, \ldots, x_{n_i}\}$ represents samples (i.e., sentences) with a token length of $n_i$, and $N_t$ denotes the number of samples. A model $\pi$ is trained on $\mathcal{D}_t$, and its optimal parameters $\boldsymbol{\theta}^*$ satisfy the following equation:

$$\boldsymbol{\theta}^* = \arg\min_{\boldsymbol{\theta}} \mathcal{L}_{\mathrm{NLL}}(\mathcal{D}_t; \boldsymbol{\theta}) = \arg\min_{\boldsymbol{\theta}} -\mathbb{E}_{\boldsymbol{x}^i \sim \mathcal{D}_t}\left[\sum_{t=1}^{n_i} \log p(x_t \mid \boldsymbol{x}_{<t}; \boldsymbol{\theta})\right], \quad (1)$$

where $p(x_t \mid \boldsymbol{x}_{<t}; \boldsymbol{\theta}) = \pi_{\boldsymbol{\theta}}(x_t \mid \boldsymbol{x}_{<t})$ denotes the prediction probability of model $\pi_{\boldsymbol{\theta}}$ for the $t$-th token, given the first $t-1$ tokens as input. Next, we define the objectives for conducting RLHF and MU on the model $\pi_{\boldsymbol{\theta}}$, respectively.

### 2.1 DEFINITION OF RLHF

The standard RLHF paradigm consists of two main stages (Azar et al., 2024): i) learning a reward model, and ii) optimizing the policy (i.e., the model parameters) based on the learned reward.

In the reward model learning phase, a binary classifier is often trained using a logistic regression loss to distinguish preferred from non-preferred behaviors. A popular choice is the Bradley-Terry model (Bradley & Terry, 1952), where the pointwise reward $r(\boldsymbol{x}_{<t}, x_t)$ serves as the score for action $x_t$, given context $\boldsymbol{x}_{<t}$. Given a dataset $\mathcal{D}_a = \{\boldsymbol{x}^i_{<t}, x^i_t \succ \hat{x}^i_t\}_{i=1}^{N_a}$, where $x^i_t \succ \hat{x}^i_t$ denotes a preference for $x^i_t$ over $\hat{x}^i_t$, the reward function is learned by minimizing the following logistic regression loss:

$$\mathcal{L}(r) = -\mathbb{E}_{(\boldsymbol{x}^i_{<t}, x^i_t \succ \hat{x}^i_t) \sim \mathcal{D}_a}\left[\log\left(p(x^i_t \succ \hat{x}^i_t | \boldsymbol{x}^i_{<t})\right)\right], \quad (2)$$

where $p(x_t^i \succ \hat{x}_t^i | \boldsymbol{x}_{<t}^i) = \sigma\left(r(\boldsymbol{x}_{<t}^i, x_t^i) - r(\boldsymbol{x}_{<t}^i, \hat{x}_t^i)\right)$ and $\sigma(\cdot)$ denotes the sigmoid function.

Based on the reward function, the objective of RLHF is to maximize the expected reward while minimizing the divergence between the policy $\pi_{\boldsymbol{\theta}}$ and a reference policy $\pi_{\mathrm{ref}}$. The specific objective can be expressed as:

$$\mathcal{J}(\boldsymbol{\theta}) = \mathbb{E}_{\pi_{\boldsymbol{\theta}}}[r(\boldsymbol{x}_{<t}^i, x_t^i)] - \tau D_{\mathrm{KL}}(\pi_{\boldsymbol{\theta}} \parallel \pi_{\mathrm{ref}}), \tag{3}$$

where $\boldsymbol{x}_{<t}^i \sim \rho$ denote the sampled history, $x_t^i \sim \pi_{\boldsymbol{\theta}}(\cdot | \boldsymbol{x}_{<t}^i)$ denote the action drawn from the policy, and $\tau$ is the parameter balancing the alignment and regularization objectives. The KL divergence $D_{\mathrm{KL}}$ is used to quantify the difference between the reference and current policies. Since LLM unlearning in this work incorporates a regularization term with analogous effects, we retain only the reward term in Eq. 3.

## 2.2 DEFINITION OF LLM UNLEARNING

Mainstream methods for unlearning in LLMs typically involve fine-tuning the original model with an unlearning objective function. Giver a forget set $\mathcal{D}_f$, while specific designs vary, the loss function in LLM unlearning tasks can generally be expressed as:

$$\mathcal{L}(\boldsymbol{\theta}) = \mathcal{L}_{\mathrm{forget}}(\mathcal{D}_f; \boldsymbol{\theta}) + \lambda \mathcal{L}_{\mathrm{reg}}(\boldsymbol{\theta}). \tag{4}$$

Here, $\mathcal{L}_{\mathrm{forget}}$ often is a loss term targeting data to be unlearned, reducing the model's performance on these samples to minimize their influence on future predictions. To preserve the model's overall performance on unrelated data and confine unlearning to the intended scope, regularization terms $\mathcal{L}_{\mathrm{reg}}$ such as output loss or divergence regularization are commonly introduced. These terms essentially act as parameter regularization. Specifically, commonly used loss-based methods (Jia et al., 2024a; Ji et al., 2024a) typically integrate one or more of the loss components. For readability, in this paper, we employ the widely adopted gradient ascent unlearning loss and parameter regularization loss as general objectives for LLM unlearning, considering their broad applicability. The formalization is as follows:

$$\min_{\boldsymbol{\theta}} \underbrace{\frac{1}{|\mathcal{D}_f|} \sum_{i=1}^{|\mathcal{D}_f|} \sum_{t=1}^{n_i} \log p(x_t \mid \boldsymbol{x}_{<t}^i; \boldsymbol{\theta})}_{\mathcal{L}_{\mathrm{forget}}(\mathcal{D}_f; \boldsymbol{\theta})} + \lambda \underbrace{\|\boldsymbol{\theta} - \boldsymbol{\theta}^*\|_p^2}_{\mathcal{L}_{\mathrm{reg}}(\boldsymbol{\theta})}. \tag{5}$$

A more detailed discussion on the definition of LLM unlearning can be found in Appendix B.

## 3 CONNECTION BETWEEN MU AND PA

### 3.1 IMPACT OF MU ON PA

Given a training sample $\boldsymbol{x}$ to be unlearned, the unlearning objective in an LLM is described by Eq 5. We adopt a special bi-level optimization framework to link MU with PA, quantifying how unlearning a single sample affects the model's PA performance. In this setup, the inner problem ensures the unlearning objective is achieved, while the outer problem evaluates its impact on PA performance. Specifically, we assume that the degree of unlearning for a sample $\boldsymbol{x}$ is represented by the weight $\boldsymbol{\omega} \geq 0$, and the model parameters that satisfy the unlearning objective under this condition are denoted as $\boldsymbol{\theta}^*(\boldsymbol{\omega})$. The bi-level optimization problem is formulated as:

$$\begin{aligned} \text{Find} \quad & \mathcal{J}(\boldsymbol{\theta}^*(\boldsymbol{\omega})) - \mathcal{J}(\boldsymbol{\theta}^*(0)) \\ \text{s.t.} \quad & \boldsymbol{\theta}^*(\boldsymbol{\omega}) = \arg \min_{\boldsymbol{\theta}} \boldsymbol{\omega} \mathcal{L}_{\mathrm{forget}}(\boldsymbol{x}; \boldsymbol{\theta}) + \lambda \mathcal{L}_{\mathrm{reg}}(\boldsymbol{\theta}), \end{aligned} \tag{6}$$

where $\mathcal{J}(\boldsymbol{\theta}^*(\boldsymbol{\omega}))$ represents the model's PA performance when unlearning weight is $\boldsymbol{\omega}$. For example, $\mathcal{J}(\boldsymbol{\theta}^*(0))$ represents the model's PA performance without unlearning. Inspired by the implicit function method for solving bi-level optimization problems, we further derive Proposition 3.1.

**Assumption 3.1.** $\mathcal{L}_{\mathrm{forget}}(\boldsymbol{x}; \boldsymbol{\theta})$ *is continuously differentiable w.r.t.* $\boldsymbol{\theta}$, *and its Hessian matrix* $\boldsymbol{H}$ *is positive semidefinite (i.e.,* $\boldsymbol{x}^\top \boldsymbol{H} \boldsymbol{x} \geq 0$, $\boldsymbol{x} \neq \boldsymbol{0}$*).* $\mathcal{J}(\boldsymbol{\theta})$ *is twice continuously differentiable w.r.t* $\boldsymbol{\theta}$.

**Proposition 3.1.** *If Assumptions 3.1 holds,* $\mathcal{L}_{\mathrm{reg}}$ *adopts the 2-norm, the change in PA performance for a model with parameters* $\boldsymbol{\theta}^*$ *after unlearning sample* $\boldsymbol{x}$ *using unlearning weight* $\boldsymbol{\omega}$ *satisfies:*

$$\Delta \mathcal{J}(\boldsymbol{\theta}^*(\boldsymbol{\omega})) \approx -\frac{\boldsymbol{\omega}}{2} \nabla_{\boldsymbol{\theta}} \mathcal{J}(\boldsymbol{\theta}^*)^\top \nabla_{\boldsymbol{\theta}} \mathcal{L}_{\mathrm{forget}}(\boldsymbol{x}; \boldsymbol{\theta}^*). \tag{7}$$

Detailed proof can be found in Appendix C.1. According to Proposition 3.1, we can directly set the unlearning weight $\boldsymbol{\omega}$ to a given positive value, such as 1 (i.e., $\Delta\mathcal{J}(\boldsymbol{\theta}^*(1))$), to quantitatively assess the impact of unlearning a single sample on the model's PA performance. To further analyze the factors influencing $\Delta\mathcal{J}(\boldsymbol{\theta}^*(\boldsymbol{\omega}))$, we decompose the gradient inner product into the gradient norm and the cosine of the angle between gradients: $\Delta\mathcal{J}(\boldsymbol{\theta}^*(\boldsymbol{\omega})) \approx -\frac{\boldsymbol{\omega}}{2}\|\nabla_{\boldsymbol{\theta}}\mathcal{J}(\boldsymbol{\theta}^*)\| \cdot \|\nabla_{\boldsymbol{\theta}}\mathcal{L}_{\text{forget}}(\boldsymbol{x};\boldsymbol{\theta}^*)\| \cdot \cos(\phi)$, where $\cos(\phi)$ denotes the angle between gradient vectors. Then, we can draw the facts:

- **Fact 1: impact can be positive or negative.** The impact of unlearning a sample on PA performance can be either positive or negative, depending on the gradient direction relationship (i.e., the sign of $\cos(\phi)$), which is partially influenced by the reward of the unlearned sample's combination. A sample $\boldsymbol{x}$ can be represented as multiple combinations, i.e., $\boldsymbol{x} = \{\boldsymbol{x}_{<t}, x_t\}_{t=1}^n$. For low-reward combinations, where generated behavior often deviates significantly from human preferences, the unlearning objective gradient direction (i.e., the direction increasing the sample's generation probability) is more likely to oppose the PA objective gradient direction. This results in $\cos(\phi) < 0$ and $\Delta\mathcal{J}(\boldsymbol{\theta}^*(\boldsymbol{\omega})) > 0$. Thus, if the rewards for most combinations $\{\boldsymbol{x}_{<t}, x_t\}$ in a sample $\boldsymbol{x}$ are low, unlearning the sample tends to improve preference alignment. Conversely, if only a few combinations have low rewards, unlearning the sample will likely hinder PA.

- **Fact 2: magnitude of impact varies.** The effect of unlearning on PA performance is sample-dependent, influenced by unlearning degree and gradient norm. The gradient norm is an inherent property of sample, such as the model's degree of fit to the sample. For samples that the model fits well, the gradient norm tends to be smaller. On the other hand, the unlearning weight is a controllable factor that can be adjusted by tuning parameters such as the unlearning weight.

### 3.2 A Weighted MU Framework for PA

The above analysis indicates that, given a set $\mathcal{D} = \{\boldsymbol{x}^i\}_{i=1}^n$ containing $n$ negative samples, simply performing the unlearning operation directly according to Eq. 5 does not guarantee optimal PA results. This is primarily due to the following two issues:

- **Issue 1.** Fact 1 suggests that for a given negative sample $\boldsymbol{x}^i$, which contains some low-reward combinations, this alone does not imply that unlearning $\boldsymbol{x}^i$ will necessarily promote PA. The effectiveness of unlearning also depends on the proportion of low-reward components within the sample. This indicates that not all negative samples need to be unlearned.

- **Issue 2.** Fact 2 indicates that even if different negative samples (e.g., $\boldsymbol{x}^i$ and $\boldsymbol{x}^j$) can both promote PA, the degree of promotion may vary. This difference can be controlled by adjusting the unlearning weight $\boldsymbol{\omega}$.

Formally, in Theorem C.2 of Appendix C.2, we theoretically prove that if there is room for improvement in the PA objective, a reweighting scheme $\boldsymbol{\omega}$ exists that can improve PA without increasing the MU task loss. This property forms the theoretical basis for our subsequent algorithm design.

**Problem setup.** To address these two issues, we propose a framework called Unlearning to Align (U2A) based on a sample-weighting approach. This framework achieves the maximization of PA performance by assigning higher weights to samples that contribute more significantly to performance improvement during the unlearning process. Specifically, when the weight $\boldsymbol{\omega}$ is set to 0, it indicates that the corresponding sample is not selected for unlearning. For ease of analysis and discussion, we assume that the weight vector $\boldsymbol{\omega} = [\boldsymbol{\omega}_1, \boldsymbol{\omega}_2, \ldots, \boldsymbol{\omega}_n]$ lies on an $n$-dimensional simplex, with each element being no less than 0, and we denote the unlearning loss of each sample $\boldsymbol{x}^i$ as $\ell_i(\boldsymbol{\theta})$. The U2A framework can be formalized as solving the following optimization problem:

$$\min_{\boldsymbol{\omega}\in\Delta_n} \quad -\mathcal{J}(\boldsymbol{\theta}^*(\boldsymbol{\omega})) + \beta\mathrm{L}_p(\boldsymbol{\omega})$$

$$\text{s.t.} \quad \boldsymbol{\theta}^*(\boldsymbol{\omega}) = \arg\min_{\boldsymbol{\theta}} \sum_{i=1}^n \boldsymbol{\omega}_i\ell_i(\boldsymbol{\theta}) + \lambda\mathcal{L}_{\text{reg}}(\boldsymbol{\theta}), \tag{8}$$

where $\mathrm{L}_p(\boldsymbol{\omega})$ represents an introduced $\mathrm{L}_p$-norm sparsity-inducing regularization term to ensure that the number of selected samples for unlearning is as small as possible, and $\beta$ denotes the weight coefficient of the regularization term. Further analysis shows that when $p = 1$, the sparsity regularization has relatively weak compressive effects on small values. On the other hand, when $p = 0$, it can effectively control the sparsity of weights (i.e., the number of non-zero weights). However, in this case, the regularization term $\mathrm{L}_q(\boldsymbol{\omega})$ becomes a non-continuous and non-convex function, which

significantly increases the difficulty of optimization. Considering these factors, when $\boldsymbol{\omega} \geq 0$ and $p = \frac{1}{2}$, the regularization term $\mathrm{L}_p(\boldsymbol{\omega})$ is both a strictly convex function and exhibits good smoothness. Therefore, in this paper, we set $p = \frac{1}{2}$, making $\mathrm{L}_p(\boldsymbol{\omega}) = \sum_{i=1}^{n} \sqrt{\boldsymbol{\omega}_i}$. Solving Eq. 13 yields the selected unlearning set $\mathcal{S}$ as well as the unlearning weight $\boldsymbol{\omega}$ for each sample.

**U2A framework.** To enhance clarity, we denote the outer objective function as $g(\boldsymbol{\omega})$ and the inner objective function as $f(\boldsymbol{\theta}, \boldsymbol{\omega})$. If $f(\boldsymbol{\theta}, \boldsymbol{\omega})$ is twice differentiable w.r.t. $\theta$, the constraint $\boldsymbol{\theta}^*(\boldsymbol{\omega}) = \arg\min_{\boldsymbol{\theta}} f(\boldsymbol{\theta}, \boldsymbol{\omega})$ can be relaxed into $\frac{\partial f(\boldsymbol{\theta}, \boldsymbol{\omega})}{\partial \boldsymbol{\theta}}|_{\boldsymbol{\theta} = \boldsymbol{\theta}^*(\boldsymbol{\omega})} = 0$. When $f(\boldsymbol{\theta}, \boldsymbol{\omega})$ is strictly convex w.r.t. $\theta$, this relaxation becomes tight (Borsos et al., 2024). Assumption 3.1 ensures this property, enabling the use of first-order optimization methods (Pedregosa, 2016; Finn et al., 2017; Liu et al., 2019) to solve Eq. 6 and avoiding computationally expensive naive greedy algorithms. Considering the efficiency, we adopt a variant of the cone-constrained generalized matching pursuit algorithm (Locatello et al., 2017), which performs incremental optimization. This approach iteratively constructs the unlearning set $\mathcal{S}$, thereby significantly reducing computational complexity.

Specifically, in each iteration, we first solve the inner optimization problem using the gradient descent method to obtain the model parameters $\boldsymbol{\theta}^*(\boldsymbol{\omega})$ with optimal unlearning performance. After completing the inner optimization, we identify a new sample point $k$ to add to the unlearning set based on the marginal gain of the outer objective function $g(\boldsymbol{\omega})$, to maximize the marginal gain. The marginal gain is calculated as $\Delta g(k) = -\frac{\partial g(\boldsymbol{\omega})}{\boldsymbol{\omega}_k}$. According to the implicit function theorem, the gradient of $g(\boldsymbol{\omega})$ w.r.t. $\boldsymbol{\omega}$ can be expressed as:

$$\Delta g(k) = -\nabla_{\boldsymbol{\theta}} \mathcal{J}(\boldsymbol{\theta}^*(\boldsymbol{\omega}))^\top \left(\frac{\partial^2 f}{\partial \boldsymbol{\theta}^2}\right)^{-1} \nabla_{\boldsymbol{\theta}} \ell_k(\boldsymbol{\theta}^*(\boldsymbol{\omega})) - \frac{\beta}{2} \boldsymbol{\omega}_k^{-\frac{1}{2}}. \tag{9}$$

where $\frac{\partial^2 f}{\partial \boldsymbol{\theta}^2} = \sum_{i=1}^{n} \boldsymbol{\omega}_i \nabla_{\boldsymbol{\theta}}^2 \ell_i(\boldsymbol{\theta}^*(\boldsymbol{\omega})) + 2\lambda I$, denoting the Hessian matrix of the inner optimization problem (details are provided in Appendix C.3). After computing the marginal gain, we select the sample point $k^*$ with the maximum gain, add it to the unlearning set $\mathcal{S}_{t-1}$.

Subsequently, we fix the model parameters to solve the outer optimization problem to obtain the solution $\boldsymbol{\omega}^{t,*}$, which can be formalized as:

$$\boldsymbol{\omega}^{t,*} = \arg\min_{\boldsymbol{\omega} \in \Delta_n} g(\boldsymbol{\omega}) \quad \text{s.t.} \quad \text{supp}(\boldsymbol{\omega}) = \mathcal{S}_t, \tag{10}$$

where the constraint is imposed to restrict the support set of the weight vector $\boldsymbol{\omega}$ to be identical to the current unlearning set $\mathcal{S}_t$. In other words, the non-zero components of $\boldsymbol{\omega}$ are confined to elements within the current unlearning set, thereby preventing the introduction of new sample points. The support set $\text{supp}(\boldsymbol{\omega}) = \{i \mid \boldsymbol{\omega}_i \neq 0\}$ denotes the indices of the non-zero entries in $\boldsymbol{\omega}$. Let $s = |\mathcal{S}_t|$ be the support set size, and $\boldsymbol{\omega}_{\mathcal{S}_t} \in \mathbb{R}^s$ the corresponding subvector. The equivalent optimization reduces to $\min_{\boldsymbol{\omega}_{\mathcal{S}_t} \in \Delta_{s-1}} g(\boldsymbol{\omega})$, subject to a simplex constraint, and can be solved by mirror descent (Wang et al., 2021; Zhang et al., 2021). The update rule is given by:

$$\boldsymbol{\omega}^{t+1} = \arg\min_{\boldsymbol{\omega} \in \Delta_{s-1}} \left\langle \nabla g(\boldsymbol{\omega}^t), \boldsymbol{\omega} \right\rangle + \frac{1}{\eta} D_h \left(\boldsymbol{\omega} \,\|\, \boldsymbol{\omega}^t\right), \tag{11}$$

where $\eta$ is the step size, $\langle \cdot, \cdot \rangle$ denotes the dot product, and $D_h(\cdot\|\cdot)$ is the Bregman divergence (Fatkhullin & He, 2024). Using the negative entropy $h(\boldsymbol{x}) = \sum \boldsymbol{x}_i \ln \boldsymbol{x}_i - \boldsymbol{x}_i$, the update has a closed form:

$$\boldsymbol{\omega}_i^{t+1} = \frac{\boldsymbol{\omega}_i^t \cdot \exp\left(-\eta \cdot \nabla_i g(\boldsymbol{\omega}^t)\right)}{\sum_{j=1}^{s} \boldsymbol{\omega}_j^t \cdot \exp\left(-\eta \cdot \nabla_j g(\boldsymbol{\omega}^t)\right)}, \quad \text{s.t.} \quad i \in \mathcal{S}_t. \tag{12}$$

This yields a smooth, probabilistic update guided by the gradient. Thus, we can obtain the complete process of U2A in Algorithm 1. In practice, to improve efficiency, U2A initialization (first point selection) typically involves multiple points controlled by a tunable hyperparameter $M$, and each update is similarly governed by $N$. More detailed implementation specifics provided in Appendix D.

**Convergence and complexity analysis.** We analyze the convergence of the U2A framework and obtain Lemma C.1 and Lemma C.2 (see Appendix C.4), which prove that it has good convergence properties. We further analyze the complexity of U2A (Appendix C.5), showing that it is computationally efficient and well-suited for high-dimensional settings.

# 4 EXPERIMENT

## 4.1 EXPERIMENT SETUPS

**Datasets and models.** We evaluate across three mainstream PA tasks and datasets: (i) reducing harmfulness **SafeRLHF** (Dai et al., 2023), (ii) enhancing usefulness **UltraFeedback** (Cui et al., 2023; Tunstall et al., 2023), (iii) eliminating hallucinations **HaluEval** (Li et al., 2023). Following the configurations in prior study (Jia et al., 2024a; Zhou et al., 2024b), we select the widely used Llama-2-7B-Chat (LLaMA2) (Touvron et al., 2023) and Llama-3.1-8B-Instruct (LLaMA3) (Dubey et al., 2024) as base models for each dataset. For PA evaluation, we employ the Skywork-Reward-Llama-3.1-8B model (Liu et al., 2024a). We report the details of training in Appendix E.1.

**Pipeline and data pre-processing.** To evaluate the effectiveness of the proposed unlearning method on preference datasets, we design a unified data processing pipeline (Figure 5(a) in Appendix F.1). An ablation study on the effects of negative ratio and forget ratio is presented in Appendix F.1. Due to the space limit, please refer to Appendix F.1.

**Evaluation metrics.** We evaluate our proposed method along two dimensions: PA performance and unlearning performance. For unlearning performance, we evaluate unlearning effectiveness and model utility. Due to space limit, details are provided in Appendix E.2.

**Baselines.** We evaluate our proposed method U2A against widely acknowledged baselines, including unlearning methods (i.e., Retrain, GA (Maini et al., 2024), GradDiff (Liu et al., 2022; Yao et al., 2024), and NPO (Zhang et al., 2024)), as well as PA methods (i.e., PPO (Schulman et al., 2017) and DPO (Rafailov et al., 2024)). The effectiveness of our method is validated by comparing the U2A-improved unlearning baseline with the original baseline and existing PA baselines.

## 4.2 EXPERIMENT RESULTS

**Unlearning affects PA.** We assess the impact of unlearning individual samples on PA performance using the LLaMA2 model across three datasets. Given the nearly negligible effect of unlearning a single sample on model parameters, we randomly select 150 groups, each with 32 negative samples, from a pool of eligible negative samples. PA performance changes after unlearning each group are compared, with parameter $\omega$ set to 1. Figure 1 shows that unlearning can have both positive and negative effects, suggesting that removing negative samples does not consistently improve PA performance. Additionally, the degree of improvement

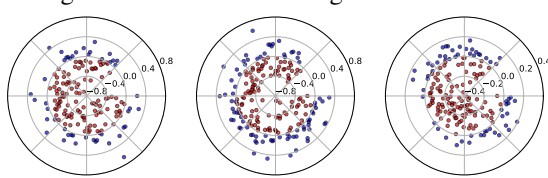

(a) SafeRLHF    (b) UltraFeedback    (c) HaluEval

Figure 1: Effect of unlearning individual data samples on PA performance of LLaMA2 model. Each point represents the PA performance change after unlearning a specific data sample. The angle of each point follows a uniform distribution, while the radial distance indicates the magnitude of PA performance change. Red points represent negative effects (i.e., unlearning this sample led to worse PA), whereas blue points represent positive effects (i.e., unlearning this sample improved PA). Note that larger distances from the origin correspond to stronger impacts on PA performance.

varies significantly across different samples. For details on the computation of PA performance, refer to Appendix E.3.

To better understand this, we decompose reward values for each token. Tokens with reward values below the average are classified as "low-reward", while those above the average are "high-reward". The average reward values for each dataset are as follows: -1.74 for SafeRLHF, 0.90 for UltraFeed-back, and -0.78 for HaluEval. To distinguish the impact of different sample groups, we apply a threshold on the proportion of low-reward tokens. Specifically, samples with a low-reward token proportion below the threshold are marked in red, while those exceeding the threshold are marked in blue. Figure 2 illustrates the impact of unlearning these samples on PA performance. Taking SafeRLHF as an example, most changes in PA performance are positive when the proportion of low-reward tokens in the unlearned dataset exceeds the threshold (dashed vertical line at 0.6). Conversely, when the proportion of low-reward tokens is below the threshold, most changes are negative. In general, red samples, with more dispersed rewards and fewer low-reward tokens, tend to hinder the improvement in PA performance when not learned. In contrast, unlearning blue samples, with a higher proportion of low-reward tokens, significantly boosts PA performance. These findings align with our theoretical analysis in Section 3.1.

**Effectiveness of U2A.** To comprehensively evaluate the effectiveness and applicability of the proposed U2A framework in enhancing PA, we conduct experiments on two widely used datasets:

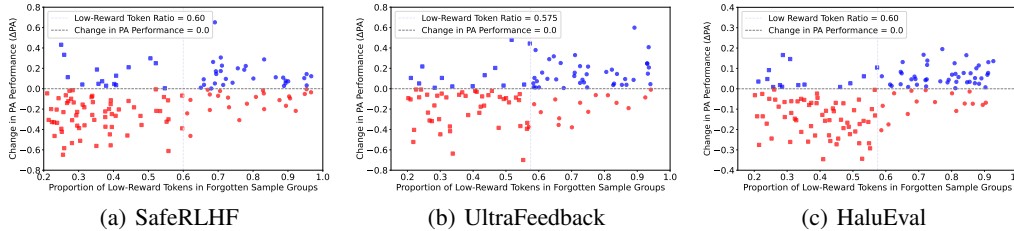

|                | (a) SafeRLHF | (b) UltraFeedback | (c) HaluEval |

Figure 2: Analysis of how unlearning data samples affects PA performance. Each point represents the change in PA performance ($\Delta$PA) after unlearning a group of samples. The $x$-axis denotes the proportion of low-reward tokens in the unlearned sample groups, and the $y$-axis represents the corresponding change in PA performance.

SafeRLHF and UltraFeedback. LLaMA2 and LLaMA3 are selected as representative open-source base models. The experimental setup comprises two parts. The first assesses different variants of the unlearning method within U2A, comparing their PA performance and unlearning performance against a baseline unlearning strategy that removes negative samples. This aims to validate the practical benefits of U2A. The second part benchmarks U2A against mainstream PA methods, demonstrating its potential and competitiveness in improving PA performance.

- **Analysis of the improvement brought by U2A.** To evaluate the enhancement achieved by U2A, we compare it against three widely used MU methods: GA, GradDiff, and NPO. All methods are applied to the complete set of negative samples, simulating the full-sample unlearning paradigm commonly adopted in traditional MU methods. A retrained model is also included as a reference, serving as the "gold standard" for full unlearning to assess the trade-off between performance and cost. On this basis, we construct three U2A-integrated variants (i.e., GA & U2A, GradDiff & U2A, and NPO & U2A) by incorporating the proposed sample weighting mechanism. This design enables evaluation of whether U2A enhances PA under the same unlearning settings. Detailed results are presented in Table 1 and Table 6 (Appendix F.2), from which we draw the following observations:

Table 1: The performance of unlearning methods on the SafeRLHF dataset before and after U2A integration. Methods outperforming their respective baselines are indicated in *italics*, and the overall best-performing method is highlighted in **bold**.

| Models | Methods | PA Performance | | | | | MU Performance | |
|--------|---------|----------------|--|--|--|--|----------------|--|
|        |         | Reward-V ($\uparrow$) | ASR-K ($\downarrow$) | ASR-A ($\downarrow$) | ASR-U ($\downarrow$) | ASR-S ($\downarrow$) | MIA ($\uparrow$) | PPL ($\downarrow$) |
| LLaMA2 | Original | -20.361 | 0.933 | 0.837 | 0.162 | 0.812 | - | 8.538 |
|        | Retrain | -18.243 | 0.873 | 0.602 | 0.177 | 0.563 | 0.521 | 8.924 |
|        | GA | -19.103 | 0.942 | 0.792 | 0.217 | 0.725 | 0.515 | 7.003 |
|        | **GA & U2A** | *-15.993* | *0.746* | *0.112* | *0.158* | *0.094* | *0.556* | 11.021 |
|        | GradDiff | -19.545 | 0.956 | 0.815 | 0.164 | 0.783 | 0.511 | 5.982 |
|        | **GradDiff & U2A** | ***-15.843*** | ***0.644*** | ***0.089*** | *0.142* | ***0.067*** | ***0.584*** | 14.573 |
|        | NPO | -20.041 | 0.927 | 0.840 | 0.121 | 0.812 | 0.504 | **5.079** |
|        | **NPO & U2A** | *-19.639* | 0.946 | *0.831* | ***0.100*** | 0.812 | *0.503* | 5.294 |
| LLaMA3 | Original | -19.827 | 0.971 | 0.848 | 0.148 | 0.794 | - | **7.627** |
|        | Retrain | -17.911 | 0.958 | 0.740 | 0.138 | 0.686 | 0.527 | 10.412 |
|        | GA | -18.011 | 0.982 | 0.829 | 0.163 | 0.750 | 0.509 | 20.405 |
|        | **GA & U2A** | ***-7.609*** | ***0.246*** | *0.162* | ***0.123*** | ***0.121*** | *0.522* | 17.876 |
|        | GradDiff | -18.477 | 0.971 | 0.829 | 0.179 | 0.767 | 0.511 | 23.426 |
|        | **GradDiff & U2A** | *-12.644* | *0.417* | ***0.103*** | *0.140* | *0.173* | ***0.536*** | 21.949 |
|        | NPO | -17.985 | 0.958 | 0.860 | 0.131 | 0.785 | 0.505 | 11.201 |
|        | **NPO & U2A** | *-13.815* | *0.783* | *0.698* | 0.233 | *0.498* | *0.508* | 20.177 |

*(i) U2A substantially improves the PA performance of all baseline methods, achieving optimal results.* While basic unlearning methods applied to all negative samples yield moderate gains, they remain inferior to retraining. In contrast, U2A variants consistently outperform retraining across multiple metrics, indicating that retraining is not the optimal unlearning strategy for PA, as not all unlearned samples equally contribute to improving PA.

*(ii) U2A maintains competitive unlearning performance while markedly enhancing PA.* It generally outperforms baselines in metrics like MIA and PPL, due to its targeted selection and reweighting of key samples. Nonetheless, in some cases, performance drops occur as U2A only removes a subset of the unlearning set, which may weaken overall unlearning strength.

- **In-depth correlation between MU and PA.** To explore the intrinsic link between MU and PA, we conduct a systematic comparison between U2A-enhanced unlearning methods and mainstream PA methods based on positive-negative sample pairs. Experiments are performed on the SafeRLHF and UltraFeedback datasets, using 2,407 and 12,227 positive-negative sample pairs, respectively, under two representative PA algorithms: DPO and PPO. Sample scales are matched with dataset partitioning. Results for UltraFeedback are shown in Table 2 and SafeRLHF are in Table 7 (Appendix F.3). Key observations are as follows:

Table 2: Comparison of the PA Performance between MU and PO on the UltraFeedback dataset.

| Model | Method | Reward-V ($\uparrow$) | LC-WR (%, $\uparrow$) | GPT4-WR (%, $\uparrow$) | Coherence ($\uparrow$) |
|-------|--------|------------|-------------|---------------|------------|
| LLaMA2 | Original | -8.578 | 11.93 | 6.96 | 0.7328 |
| | Retrain | -7.739 | 15.29 | 9.20 | 0.7324 |
| | PPO | -6.971 | 17.13 | 10.29 | 0.7375 |
| | DPO | **-6.728** | 16.62 | **10.82** | 0.7370 |
| | **GA** & **U2A** | -7.154 | **20.12** | 10.63 | 0.7077 |
| | **GradDiff** & **U2A** | -7.233 | 18.35 | 9.95 | 0.7091 |
| | **NPO** & **U2A** | -8.037 | 14.44 | 7.84 | **0.7380** |
| LLaMA3 | Original | -4.996 | 10.57 | 5.85 | 0.7263 |
| | Retrain | -3.318 | 15.48 | 8.93 | 0.7300 |
| | PPO | -1.302 | 23.43 | 19.41 | 0.7395 |
| | DPO | -0.277 | **29.30** | 19.08 | 0.7375 |
| | **GA** & **U2A** | 1.105 | 24.79 | **21.33** | **0.7450** |
| | **GradDiff** & **U2A** | **1.248** | 24.18 | 19.67 | 0.7380 |
| | **NPO** & **U2A** | -0.248 | 20.56 | 13.20 | 0.7294 |

*(i) MU is an effective way for PA.* On LLaMA3 and UltraFeedback, U2A achieves comparable or superior PA performance relative to DPO and PPO. This indicates that unlearning, viewed as negative sample exclusion, can serve as an effective and complementary PA strategy.

*(ii) Knowing "what to unlearn" is insufficient; learning "what to generate" is equally critical.* On the LLaMA2 model and SafeRLHF dataset, U2A shows no significant advantage over mainstream PA methods. This is because, although U2A effectively removes negative samples through a "negative avoidance" strategy, which suppresses low-quality responses to indirectly enhance PA, it still lacks explicit guidance for generating high-reward, human-aligned content. Without positive sample supervision, its ability to learn and generalize high-quality outputs is limited, reducing its efficacy compared to PA methods. This highlights the need for unlearning methods to address both undesired outputs and desired generation behaviors.

*(iii) Model generalization and negative sample quality critically affect U2A.* U2A performs better on LLaMA3 and UltraFeedback than on LLaMA2 and SafeRLHF, likely due to two factors. First, LLaMA3's larger-scale pretraining confers stronger representational and generalization capabilities, enabling the model to infer positive behaviors and avoid harmful ones even when only negative samples are removed. In contrast, LLaMA2's weaker generalization may lead to uncertainty or degraded behavior post-unlearning, impairing PA performance. Second, the UltraFeedback dataset provides high-quality negative samples that are well-annotated, semantically coherent, and distributionally focused, thereby supporting more effective behavior suppression. In contrast, the noisier and less consistent negative samples in SafeRLHF limit their impact on PA.

*(iv) The quantitative relationship between negative sample scale and PA training requirements warrants further study.* As higher PA performance is pursued, both PA methods and unlearning methods typically demand more positive-negative sample pairs for the former and negative samples for the latter. Whether a functional relationship exists between the data requirements of different methods to achieve comparable alignment levels remains an open question. Clarifying this could offer a theoretical basis for more efficient training resource allocation.

**Sensitivity analysis of hyperparameter $M$.** We investigate the effect of the hyperparameter $M$, which represents the number of samples selected for weighted unlearning at the initial step, on PA in the U2A framework, using the SafeRLHF and UltraFeedback datasets with LLaMA2 and LLaMA3 models. For different values of $M$, we select the Top-$M$ high-gain samples and apply U2A weighted unlearning. After each unlearning step, we evaluate PA performance using average reward as the main metric. The results in Figure 3 (top row) show that PA performance is generally higher for small values of $M$. When $M$ exceeds a certain threshold, performance stabilizes or decreases. To analyze this, we compute the marginal gain on PA performance for each negative sample in the candidate forget set, representing the estimated reward improvement if the sample is forgotten. The marginal-gain distributions (Figure 3, bottom row) show that a few high-gain samples

are concentrated at the left tail, while most samples have marginal gains near zero. This suggests that high-gain samples mainly improve PA performance, while low or near-zero gain samples offer little benefit and may even cause a slight degradation in PA when $M$ exceeds the threshold.

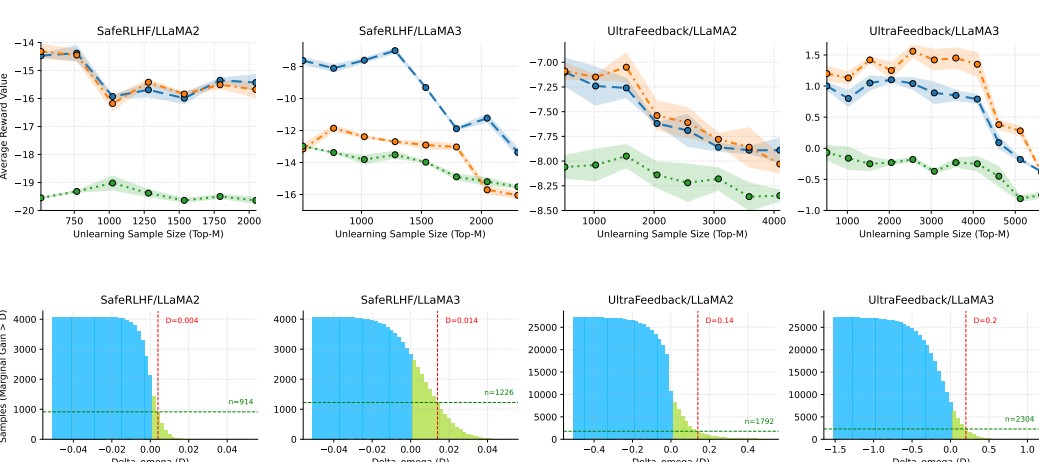

Figure 3: Sensitivity analysis of the hyperparameter $M$. As observed, unlearning high-marginal-gain samples at the early stage yields substantial improvements in preference alignment, whereas including more low-marginal-gain samples leads to saturated or even slightly degraded gains.

**Ablation study on sample selection based on marginal gain.** We conduct an ablation study on the UltraFeedback dataset with the LLaMA3 model to evaluate whether the marginal-gain-based sample selection improves PA performance. For various unlearning set sizes $K$, we compare two strategies: (i) selecting the Top-K samples with the highest marginal gains, and (ii) randomly selecting $K$ samples as a baseline. We apply the same U2A weighted unlearning procedure to both sets and evaluate PA performance using average reward. As shown in Figure 4, the models using marginal gain selection consistently outperformed those with random selection in average reward, with this advantage increasing as $K$ grew. These results confirm that marginal gain computation effectively identifies key samples for improving preference alignment, significantly enhancing U2A unlearning performance, and validating the necessity of this module.

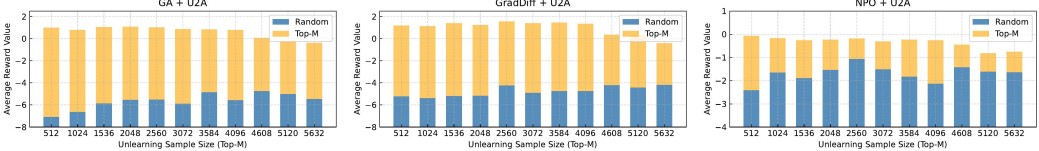

Figure 4: Ablation study. Each subfigure presents the comparison of Top-$K$ selection guided by marginal gain (yellow) and random selection (blue) in terms of improvement in PA performance.

For more experimental results (e.g., the processing efficiency of U2A), see Appendix F.

## 5 CONCLUSION

The mainstream PA method, RLHF, faces significant challenges in low-resource settings, including i) reliance on numerous positive preference samples, which are costly to obtain, and ii) instability during training, resulting in high computational costs. To address these issues, we propose a MU-based method that reduces dependence on positive samples by unlearning negative samples to achieve PA. Our method achieves computational efficiency comparable to standard fine-tuning while showing strong potential. We first develop a bi-level optimization framework to evaluate the impact of unlearning individual samples on PA performance. Through our analysis, we observe that negative samples contribute unevenly to PA, with many offering limited benefits. This observation leads to a key question: how can we selectively weight and unlearn negative samples to optimize alignment? To this end, we formally define the problem and introduce U2A, a framework leveraging bi-level optimization to efficiently select and weighted unlearn samples for improved alignment. Experiments demonstrate that U2A significantly enhances alignment efficiency and effectiveness, underscoring its value in resource-constrained scenarios. By linking PA with MU, this work provides a novel perspective on PA for LLMs and suggests new directions for optimizing PA algorithms.

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

## A  RELATED WORK

### A.1  PREFERENCE ALIGNMENT

PA methods can be broadly classified into learning-based and decoding-based methods, depending on whether model parameters are updated (Zhou et al., 2024b). Learning-based methods (Ziegler et al., 2019; Stiennon et al., 2020; Ouyang et al., 2022; Azar et al., 2024), such as RLHF, optimize models using preference datasets with techniques like PPO (Schulman et al., 2017), DPO (Rafailov et al., 2024), WPO (Zhou et al., 2024a), and Self-play Preference Optimization (SPO) (Wu et al., 2024). However, RLHF is computationally expensive (Rafailov et al., 2024). To mitigate this, decoding-based methods (Kim et al., 2023; Gao et al., 2023; Huang et al., 2024; Mudgal et al., 2024), which guide inference without parameter updates, have gained attention. Examples include rejection sampling (Mitchell et al., 2024; Beirami et al., 2024) and Monte Carlo Tree Search (Liu et al., 2023; Wan et al., 2024), which reduce computational costs by keeping parameters fixed. Since this study focuses on the relationship between MU and PA, and the former requires parameter updates, we primarily consider learning-based methods.

### A.2  LLM UNLEARNING

The goal of LLM unlearning is to remove specific knowledge from training data while preserving the model's performance on unrelated tasks (Jang et al., 2023; Ji et al., 2024a; Liu et al., 2024b; Feng et al., 2024). Existing methods can be categorized into three main approaches: i) Gradient-based methods (Jang et al., 2023; Maini et al., 2024) use gradient ascent on the forget set (i.e., the data to be unlearned) to remove associated knowledge, with parameter regularization added to preserve performance on other tasks. ii) Preference optimization-based methods (Maini et al., 2024; Zhang et al., 2024) treat the forget set as negative examples or assign predefined responses (e.g., rejection responses) to achieve unlearning during PA. iii) Model weight-based methods (Jia et al., 2024b) analyze the roles of different model modules to guide unlearning, leveraging the modularity of LLMs. As model weight-based methods are primarily used for attribution analysis, this study focuses on gradient-based and preference optimization-based approaches.

# B   DISCUSSION ON THE DEFINITION OF UNLEARNING IN LLMS

In this section, we review and summarize the definitions of existing unlearning methods for LLMs and attempt to incorporate these methods into a unified theoretical framework. Assume the training dataset is denoted as $\mathcal{D}_t = \mathcal{D}_f \cup \mathcal{D}_r$, where $\mathcal{D}_f$ represents the set of samples to be unlearned, and $\mathcal{D}_r$ represents the remaining samples. The core objective of LLM unlearning is to remove the knowledge learned from $\mathcal{D}_f$ while preserving the model's other capabilities as much as possible. To achieve this goal, existing methods can be broadly categorized into two main classes: (i) gradient-based methods and (ii) preference optimization-based methods.

**Gradient-based methods.**   Gradient-based methods include gradient ascent and its various extensions. Below, we will review the definitions of these methods sequentially.

*Gradient ascent.* Gradient ascent (Jang et al., 2023; Feng et al., 2024) is a traditional and straightforward baseline method that removes the model's memory of the samples in $\mathcal{D}_f$ by maximizing the loss on $\mathcal{D}_f$, effectively reversing the gradient descent process. It is defined as:

$$\mathcal{L}_{\text{GA}} = \underbrace{\frac{1}{|\mathcal{D}_f|} \sum_{i=1}^{|\mathcal{D}_f|} \sum_{t=1}^{n_i} \log p(x_t \mid \boldsymbol{x}_{<t}; \boldsymbol{\theta})}_{\mathcal{L}_{\text{forget}}(\mathcal{D}_f; \boldsymbol{\theta})},$$

where $n_i$ denotes the number of tokens in the sample $\boldsymbol{x}^i$, and $\boldsymbol{\theta}$ represents the parameters of the model.

*Variants of gradient ascent.* However, naive gradient ascent significantly degrades the model's other capabilities. To address this issue, recent studies (Wang et al., 2023; Li et al., 2024; Maini et al., 2024; Ji et al., 2024a; Jia et al., 2024b; Liu et al., 2024b; Wang et al., 2024) have introduced various regularization terms, primarily including loss-based regularization and divergence-based regularization, as described below:

- **Loss-based regularization.** Loss-based regularization (Li et al., 2024; Maini et al., 2024; Jia et al., 2024b) maintains the model's other capabilities by sampling a dataset $\mathcal{D}_r'$ that shares the same distribution as $\mathcal{D}_r$ and minimizing the model's loss on $\mathcal{D}_r'$. The formal expression is:

$$\mathcal{L}_{\text{GA+LR}} = \underbrace{\mathcal{L}_{\text{GA}}}_{\mathcal{L}_{\text{forget}}(\mathcal{D}_f; \boldsymbol{\theta})} - \lambda \underbrace{\frac{1}{|\mathcal{D}_r'|} \sum_{i=1}^{|\mathcal{D}_r'|} \sum_{t=1}^{n_i} \log p(x_t \mid \boldsymbol{x}_{<t}; \boldsymbol{\theta})}_{\mathcal{L}_{\text{reg}}(\boldsymbol{\theta})}.$$

- **Divergence-based regularization.** Similar to loss-based regularization, divergence-based regularization (Wang et al., 2023; Tian et al., 2024; Maini et al., 2024; Ji et al., 2024a; Wang et al., 2024) preserves model performance by constraining the output distribution of the model on a dataset $\mathcal{D}_r'$. Specifically, this method minimizes the distributional distance $Dis(\cdot \parallel \cdot)$ between the output distribution of the unlearned model on $\mathcal{D}_r'$ and that of the original model on $\mathcal{D}_r'$. Depending on the metric used to measure the distributional distance, this method can further be categorized into regularizations based on KL divergence (Wang et al., 2023; Maini et al., 2024) and f-divergence (Wang et al., 2024). The formal definition is:

$$\mathcal{L}_{\text{GA+DR}} = \underbrace{\mathcal{L}_{\text{GA}}}_{\mathcal{L}_{\text{forget}}(\mathcal{D}_f; \boldsymbol{\theta})} + \lambda \underbrace{\frac{1}{|\mathcal{D}_r'|} \sum_{i=1}^{|\mathcal{D}_r'|} \sum_{t=1}^{n_i} Dis(P(\cdot \mid \boldsymbol{x}_{<t}; \boldsymbol{\theta}) \parallel P(\cdot \mid \boldsymbol{x}_{<t}; \boldsymbol{\theta}^*))}_{\mathcal{L}_{\text{reg}}(\boldsymbol{\theta})}.$$

Both loss regularization and divergence regularization can essentially be regarded as forms of parameter regularization, which constrain the norm of the difference between model parameters before and after unlearning to be less than a threshold $\delta$. By restricting the parameter changes within a $\delta$-norm ball, this technique ensures the preservation of the model's other capabilities. However, parameter regularization is difficult to handle directly as a constraint, so its relaxed form is often utilized and

incorporated into the objective function instead. Formally, this can be expressed as:

$$\mathcal{L}_{\text{GA+PR}} = \underbrace{\mathcal{L}_{\text{GA}}}_{\mathcal{L}_{\text{forget}}(\mathcal{D}_f;\boldsymbol{\theta})} + \lambda \underbrace{\|\boldsymbol{\theta} - \boldsymbol{\theta}^*\|_p^2}_{\mathcal{L}_{\text{reg}}(\boldsymbol{\theta})}.$$

In addition, while several recent methods (i.e., Mismatch and LLMU) (Yao et al., 2024) differ in their definitions of unlearning objectives, they are fundamentally variants of gradient ascent methods. These methods further refine gradient ascent by extending the formulation of unlearning objectives, constructing a random combination of text sequences $Y_{\text{ran}}$. Specifically, their definitions are given as:

$$\mathcal{L}_{\text{MIS}} = \underbrace{-\frac{1}{|\mathcal{D}_f|}\sum_{i=1}^{|\mathcal{D}_f|}\sum_{t=1}^{n_i}\frac{1}{|Y_{\text{ran}}|}\sum_{j=1}^{|Y_{\text{ran}}|}\log p(y_{\text{ran}}^j \mid \boldsymbol{x}_{<t};\boldsymbol{\theta})}_{\mathcal{L}_{\text{forget}}(\mathcal{D}_f;\boldsymbol{\theta})} - \lambda \underbrace{\frac{1}{|\mathcal{D}_r'|}\sum_{i=1}^{|\mathcal{D}_r'|}\sum_{t=1}^{n_i}\log p(x_t \mid \boldsymbol{x}_{<t};\boldsymbol{\theta})}_{\mathcal{L}_{\text{reg}}(\boldsymbol{\theta})}.$$

$$\mathcal{L}_{\text{LLMU}} = \mathcal{L}_{\text{GA}} \underbrace{- \frac{1}{|\mathcal{D}_f|}\sum_{i=1}^{|\mathcal{D}_f|}\sum_{t=1}^{n_i}\frac{1}{|Y_{\text{ran}}|}\sum_{j=1}^{|Y_{\text{ran}}|}\log p(y_{\text{ran}}^j \mid \boldsymbol{x}_{<t};\boldsymbol{\theta})}_{\mathcal{L}_{\text{forget}}(\mathcal{D}_f;\boldsymbol{\theta})}$$

$$+ \lambda \underbrace{\frac{1}{|\mathcal{D}_r'|}\sum_{i=1}^{|\mathcal{D}_r'|}\sum_{t=1}^{n_i} Dis(P(\cdot \mid \boldsymbol{x}_{<t};\boldsymbol{\theta}) \,\|\, P(\cdot \mid \boldsymbol{x}_{<t};\boldsymbol{\theta}^*))}_{\mathcal{L}_{\text{reg}}(\boldsymbol{\theta})}.$$

**Preference optimization-based methods.** Preference optimization-based unlearning methods for LLMs primarily include DPO (Lu et al., 2022; Maini et al., 2024; Dige et al., 2024) and its variants, as well as Negative Preference Optimization (NPO) (Zhang et al., 2024; Fan et al., 2024). These methods achieve unlearning by constructing additional preference data pairs and leveraging existing preference optimization algorithms to guide the model.

*DPO method.* The DPO method constructs preference data pairs based on the unlearning sample set (Lu et al., 2022; Maini et al., 2024; Dige et al., 2024). For example, given a sample $\boldsymbol{x}^i$ containing $n_i$ combinations, for any combination pair $(\boldsymbol{x}_{<t}^i, x_t)$, where $x_t$ is the truthful response, DPO sets $x_t'$ as "refuse to answer" and treats it as the preferred response. By optimizing this preference pair, DPO employs preference optimization algorithms to achieve unlearning. It is formally defined as:

$$\mathcal{L}_{\text{DPO}} = -\frac{2}{\beta}\mathbb{E}_{\boldsymbol{x}^i \in \mathcal{D}_f}\left[\log \sigma\left(\underbrace{-\beta\sum_{i=1}^{n_i}\log p(x_t \mid \boldsymbol{x}_{<t};\boldsymbol{\theta})}_{\mathcal{L}_{\text{forget}}(\mathcal{D}_f;\boldsymbol{\theta})} + \underbrace{\beta\sum_{i=1}^{n_i}\log p(x_t' \mid \boldsymbol{x}_{<t};\boldsymbol{\theta}) - M_{\text{ref}}}_{\mathcal{L}_{\text{reg}}(\boldsymbol{\theta})}\right)\right].$$

*NPO method.* The NPO method directly treats the unlearning samples as negative samples and penalizes the model's responses on the unlearning set $\mathcal{D}_f$ (Zhang et al., 2024; Fan et al., 2024). The formal definition is:

$$\mathcal{L}_{\text{NPO}} = -\frac{2}{\beta}\mathbb{E}_{\boldsymbol{x}^i \in \mathcal{D}_f}\left[\log \sigma\left(\underbrace{-\beta P(x_t \mid \boldsymbol{x}_{<t}^i;\boldsymbol{\theta}^*)}_{\mathcal{L}_{\text{forget}}(\text{forget};\boldsymbol{\theta})} + \underbrace{\beta \log P(x_t \mid \boldsymbol{x}_{<t}^i;\boldsymbol{\theta})}_{\mathcal{L}_{\text{reg}}(\boldsymbol{\theta})}\right)\right]$$

In summary, both gradient-based methods and preference optimization-based methods can be viewed as combinations of unlearning loss and regularization loss.

# C THEORETICAL ANALYSIS

## C.1 PROOF OF PROPOSITION 3.1

To analyze the variation of $\mathcal{J}(\boldsymbol{\theta}^*(\epsilon))$, we perform a Taylor expansion of Eq. 6 around $\epsilon = 0$. Here, since we are more concerned with the description of the magnitude of the effect rather than the exact values, we expand it only to the first-order term, yielding:

$$\mathcal{J}(\boldsymbol{\theta}^*(\epsilon)) \approx \mathcal{J}(\boldsymbol{\theta}^*(0)) + \epsilon \frac{\partial \mathcal{J}(\boldsymbol{\theta}^*(\epsilon))}{\partial \epsilon}|_{\epsilon=0},$$

where $\mathcal{J}(\boldsymbol{\theta}^*(0))$ represents the preference alignment performance of the model at $\epsilon = 0$ (i.e., without unlearning), while $\frac{\partial \mathcal{J}(\boldsymbol{\theta}^*(\epsilon))}{\partial \epsilon}$ denotes the rate of change in the PA performance with respect to the unlearning weight control parameter $\epsilon$. According to the chain rule, the partial derivative can be decomposed as:

$$\frac{\partial \mathcal{J}(\boldsymbol{\theta}^*(\epsilon))}{\partial \epsilon} = \nabla_{\boldsymbol{\theta}} \mathcal{J}(\boldsymbol{\theta}^*(\epsilon))^{\top} \frac{\partial \boldsymbol{\theta}^*(\epsilon)}{\partial \epsilon}.$$

Since the optimal solution $\boldsymbol{\theta}^*(\epsilon)$ of the lower-level problem satisfies the first-order optimality condition:

$$\nabla_{\boldsymbol{\theta}} \left( \epsilon \mathcal{L}_{\text{forget}}(\boldsymbol{x}; \boldsymbol{\theta}) + \mathcal{L}_{\text{reg}}(\boldsymbol{\theta}) \right)|_{\boldsymbol{\theta} = \boldsymbol{\theta}^*(\epsilon)} = 0.$$

By differentiating the above optimality condition with respect to $\epsilon$, we obtain:

$$\epsilon \nabla_{\boldsymbol{\theta}}^2 \mathcal{L}_{\text{forget}}(\boldsymbol{x}; \boldsymbol{\theta}^*(\epsilon)) \frac{\partial \boldsymbol{\theta}^*(\epsilon)}{\partial \epsilon} + \nabla_{\boldsymbol{\theta}} \mathcal{L}_{\text{forget}}(\boldsymbol{x}; \boldsymbol{\theta}^*(\epsilon)) + \nabla_{\boldsymbol{\theta}}^2 \mathcal{L}_{\text{reg}}(\boldsymbol{\theta}^*(\epsilon)) \frac{\partial \boldsymbol{\theta}^*(\epsilon)}{\partial \epsilon} = 0.$$

Substituting $\mathcal{L}_{\text{reg}}(\boldsymbol{\theta}) = \|\boldsymbol{\theta} - \boldsymbol{\theta}^*\|^2$, we have:

$$\nabla_{\boldsymbol{\theta}} \mathcal{L}_{\text{reg}}(\boldsymbol{\theta}^*(\epsilon)) = 2(\boldsymbol{\theta}^*(\epsilon) - \boldsymbol{\theta}^*), \quad \nabla_{\boldsymbol{\theta}}^2 \mathcal{L}_{\text{reg}}(\boldsymbol{\theta}^*(\epsilon)) = 2I.$$

Therefore, the implicit gradient formula is given by:

$$\frac{\partial \boldsymbol{\theta}^*(\epsilon)}{\partial \epsilon} = - \left[ \epsilon \nabla_{\boldsymbol{\theta}}^2 \mathcal{L}_{\text{forget}}(\boldsymbol{x}; \boldsymbol{\theta}^*(\epsilon)) + 2I \right]^{-1} \nabla_{\boldsymbol{\theta}} \mathcal{L}_{\text{forget}}(\boldsymbol{x}; \boldsymbol{\theta}^*(\epsilon)).$$

When $\epsilon = 0$, the formula simplifies to:

$$\frac{\partial \boldsymbol{\theta}^*(\epsilon)}{\partial \epsilon}|_{\epsilon=0} = - [2I]^{-1} \nabla_{\boldsymbol{\theta}} \mathcal{L}_{\text{forget}}(\boldsymbol{x}; \boldsymbol{\theta}^*(0)) = -\frac{1}{2} \nabla_{\boldsymbol{\theta}} \mathcal{L}_{\text{forget}}(\boldsymbol{x}; \boldsymbol{\theta}^*).$$

Substituting $\frac{\partial \boldsymbol{\theta}^*(\epsilon)}{\partial \epsilon}$ into the chain rule formula:

$$\frac{\partial \mathcal{J}(\boldsymbol{\theta}^*(\epsilon))}{\partial \epsilon}|_{\epsilon=0} = \nabla_{\boldsymbol{\theta}} \mathcal{J}(\boldsymbol{\theta}^*)^{\top} \left( -\frac{1}{2} \nabla_{\boldsymbol{\theta}} \mathcal{L}_{\text{forget}}(\boldsymbol{x}; \boldsymbol{\theta}^*) \right).$$

Therefore, the variation in preference alignment performance is:

$$\mathcal{J}(\boldsymbol{\theta}^*(\epsilon)) - \mathcal{J}(\boldsymbol{\theta}^*(0)) \approx -\frac{\epsilon}{2} \nabla_{\boldsymbol{\theta}} \mathcal{J}(\boldsymbol{\theta}^*)^{\top} \nabla_{\boldsymbol{\theta}} \mathcal{L}_{\text{forget}}(\boldsymbol{x}; \boldsymbol{\theta}^*).$$

## C.2 THE IMPACT OF SAMPLE REWEIGHTING IN BI-LEVEL OPTIMIZATION

To deepen the understanding of how sample reweighting impacts both preference alignment and MU tasks, we analyze the problem within the bi-level optimization framework:

$$\min_{\boldsymbol{\omega}:\|\boldsymbol{\omega}\|_1=1} -\mathcal{J}(\boldsymbol{\theta}^*(\boldsymbol{\omega})),$$

$$\text{s.t.} \quad \boldsymbol{\theta}^*(\boldsymbol{\omega}) = \arg\min_{\boldsymbol{\theta}} \sum_{i=1}^{n} \boldsymbol{\omega}_i \ell_i(\boldsymbol{\theta}) + \lambda \mathcal{L}_{\text{reg}}(\boldsymbol{\theta}). \tag{13}$$

We define inner unlearning objective $f(\boldsymbol{\theta}) := \sum_{i=1}^{n} \boldsymbol{\omega}_i \ell_i(\boldsymbol{\theta}) + \lambda \mathcal{L}_{\text{reg}}(\boldsymbol{\theta})$. To analyze the properties of the formulation above, we introduce the following Hessian-free assumption.

**Assumption C.1.** *(Hessian-free). The inner loss $\ell(\boldsymbol{\theta})$ is twice differentiable with respect to $\boldsymbol{\theta}$, and the Hessian matrix $\mathbf{H}(\boldsymbol{\theta}) := \sum_{i=1}^{n} \nabla_{\boldsymbol{\theta}}^2 \ell_i(\boldsymbol{\theta}) \approx \mathbf{0}$.*

This assumption is well supported by the literature on Hessian properties in neural networks. First, prior studies have established that a large fraction of the Hessian matrix's spectrum is concentrated near zero (Ghorbani et al., 2019; Sagun et al., 2016), which justifies Hessian-free or low-rank curvature approximations. Second, since ReLU networks are piecewise linear functions, their second derivatives are zero almost everywhere (Torop et al., 2023; Dauphin et al., 2024), further corroborating the assumption. Moreover, this assumption has been widely adopted in the theoretical analysis of bi-level optimization (Sow et al., 2022; Yang et al., 2023; Kornowski et al., 2024; Liu et al., 2021).

Denoting that $\mathbf{D} := \begin{bmatrix} \nabla_\theta \ell_1(\boldsymbol{\theta}^*(\boldsymbol{\omega})) \\ \vdots \\ \nabla_\theta \ell_n(\boldsymbol{\theta}^*(\boldsymbol{\omega})) \end{bmatrix} \in \mathbb{R}^{n \times m}$, we have the following property of the bi-level optimization system.

**Theorem C.2.** *Under Assumption C.1, given the sample weight $\boldsymbol{\omega} \in \mathbb{R}^n$ and the corresponding parameter $\boldsymbol{\theta}^*(\boldsymbol{\omega})$, with the constant $t > 0$ and the optimized direction $\boldsymbol{\tau} \in \mathbb{R}^n$, the impact of the sample reweighting $\boldsymbol{\omega}' := \frac{\boldsymbol{\omega} + t\boldsymbol{\tau}}{\|\boldsymbol{\omega} + t\boldsymbol{\tau}\|_1}$ with respect to PA and MU tasks can be formulated as*

$$\mathcal{I}_\mathcal{J}(\boldsymbol{\tau}; \boldsymbol{\omega}) = -\left[\nabla_\theta \mathcal{J}(\boldsymbol{\theta}^*(\boldsymbol{\omega}))^\top \mathbf{G}^{-1}(\boldsymbol{\theta}^*(\boldsymbol{\omega}))\mathbf{D}^\top\right]\boldsymbol{\tau},$$

$$\mathcal{I}_{\text{forget}}(\boldsymbol{\tau}; \boldsymbol{\omega}) = -\left[\frac{1}{n}\sum_{i=1}^n [\nabla_\theta \ell_i(\boldsymbol{\theta}^*(\boldsymbol{\omega}))]^\top \mathbf{G}^{-1}(\boldsymbol{\theta}^*(\boldsymbol{\omega}))\mathbf{D}^\top\right]\boldsymbol{\tau}.$$

*If $\sum_{i=1}^n [\nabla_\theta \ell_i(\boldsymbol{\theta}^*(\boldsymbol{\omega}))] \in \mathbb{R}^m$ and $\nabla_\theta \mathcal{J}(\boldsymbol{\theta}^*(\boldsymbol{\omega})) \in \mathbb{R}^m$ are linearly independent in the quotient space $\mathbb{R}^m \backslash \ker(\mathbf{D})$, there exists a direction $\boldsymbol{\tau}$ such that the reweighing induces a non-negative impact on both the PA and MU tasks. Furthermore, if $\mathcal{J}(\boldsymbol{\theta}^*(\boldsymbol{\omega}))$ is not in a local optimum with respect to $\boldsymbol{\omega}$, then there exists a reweighting that yields a positive impact on PA while maintaining a non-negative impact on the MU task.*

Theorem C.2 states that if the preference-alignment objective remains a space for improvement, there exists a reweighting of $\boldsymbol{\omega}$ that increases the PA objective $\boldsymbol{J}$ without increasing the unlearning loss, thereby not sacrificing the MU performance. This property provides a theoretically sound foundation for our subsequent algorithmic design.

The conditions in Theorem C.2 are mild. First, the condition that $\sum_{i=1}^n \nabla_\theta \ell_i(\boldsymbol{\theta}^*(\boldsymbol{\omega}))$ and $\nabla_\theta \mathcal{J}(\boldsymbol{\theta}^*(\boldsymbol{\omega}))$ are linearly independent in the quotient space $\mathbb{R}^m \backslash \ker(\mathbf{D})$ simply states that the gradient directions of the alignment and unlearning objectives do not coincide (nor point in exactly opposite directions) when projected onto the non-degenerate subspace defined by $\mathbf{D}$. Under stochastic gradient descent, the probability that these two projected gradients align perfectly is extremely small, making the condition mild in practice. Second, by the definition of bi-level optimization, if $\mathcal{J}(\boldsymbol{\theta}^*(\boldsymbol{\omega}))$ is already at a local optimum with respect to $\boldsymbol{\theta}$, then the bi-level system satisfies the first-order optimality condition at the corresponding $\boldsymbol{\omega}$, i.e.,

$$\frac{\partial}{\partial \boldsymbol{\omega}} \mathcal{J}(\boldsymbol{\theta}^*(\boldsymbol{\omega})) = 0,$$

and no further reweighting is necessary, ensuring the soundness of the condition.

**Proof of Theorem C.2.** We first formulate the impact of tuning the sample weight $\boldsymbol{\omega}$. For any vector $\tau \in \mathbb{R}^n$, following the prior work on influence function (Koh & Liang, 2017), the impact of reweighting $\boldsymbol{\omega}' = \boldsymbol{\omega} + t\boldsymbol{\tau}$ to the PA target $\mathcal{J}$ can be quantified by the corresponding directional derivative with respect to $t$,

$$\mathcal{I}_\mathcal{J}(\boldsymbol{\tau}; \boldsymbol{\omega}) := \frac{\partial}{\partial t} \mathcal{J}(\boldsymbol{\theta}^*(\boldsymbol{\omega} + t\boldsymbol{\tau})) = \nabla_\theta \mathcal{J}(\boldsymbol{\theta}^*(\boldsymbol{\omega}))^\top \left[\frac{\partial}{\partial t}\boldsymbol{\theta}^*(\boldsymbol{\omega} + t\boldsymbol{\tau})|_{t=0}\right]. \quad (14)$$

To proceed with the derivation, we consider the Hessian matrix of the inner optimization problem, which can be written as $\mathbf{G}(\boldsymbol{\theta}) := \mathbf{H}(\boldsymbol{\theta}) + \lambda\mathbf{I}$. Under Assumption C.1, when $\lambda > 0$, the matrix $\mathbf{G}(\boldsymbol{\theta})$ is positive definite, and its inverse exists. The first-order optimality condition of the inner optimization problem shows that

$$\nabla_\theta f(\boldsymbol{\theta}_t) = \sum_{i=1}^{n}(\boldsymbol{\omega}_i + t\boldsymbol{\tau}_i)\nabla_\theta \ell_i(\boldsymbol{\theta}_t) + \lambda \nabla_\theta \mathcal{L}_{\text{reg}}(\boldsymbol{\theta}_t) = 0. \tag{15}$$

By differentiating both sides with respect to $t$ and invoking the chain rule, we obtain

$$0 = \frac{\partial}{\partial t}\nabla_\theta f(\boldsymbol{\theta}_t) = \underbrace{\nabla_\theta^2 f(\boldsymbol{\theta}_t)}_{=:\mathbf{G}_t}\frac{\partial \boldsymbol{\theta}_t}{\partial t} + \sum_{i=1}^{n}\tau_i \nabla \ell_i(\boldsymbol{\theta}_t). \tag{16}$$

With Assumption C.1 ensuring the invertibility of $\mathbf{G}$, the derivative of $\boldsymbol{\theta}^*$ with respect to $t$ can be written as:

$$\frac{\partial}{\partial t}\theta^*(\omega + t\tau) = \frac{\partial \theta_t}{\partial t} = -\mathbf{G}_t^{-1}\left(\sum_{i=1}^{n}\tau_i \nabla \ell_i(\theta_t)\right). \tag{17}$$

Substituting $t = 0$, we obtain a simplified expression for the impact of reweighting

$$\mathcal{I}_{\mathcal{J}}(\boldsymbol{\tau}; \boldsymbol{\omega}) = -\nabla_\theta \mathcal{J}(\boldsymbol{\theta}^*(\boldsymbol{\omega}))^\top \mathbf{G}^{-1}(\boldsymbol{\theta}^*(\boldsymbol{\omega}))\left[\sum_{i=1}^{n}\boldsymbol{\tau}_i \nabla_\theta \ell_i(\boldsymbol{\theta}^*(\boldsymbol{\omega}))\right]. \tag{18}$$

By applying an analogous derivation, we can characterize the impact of reweighting on the unlearning objective $\mathcal{L}_{\text{forget}}(\boldsymbol{\theta}) := \frac{1}{n}\sum_{i=1}^{n}\ell_i(\boldsymbol{\theta})$,

$$\mathcal{I}_{\text{forget}}(\boldsymbol{\tau}; \boldsymbol{\omega}) := \frac{\partial}{\partial t}\mathcal{L}_{\text{forget}}(\boldsymbol{\theta}^*(\boldsymbol{\omega} + t\boldsymbol{\tau})) \tag{19}$$

$$= -\frac{1}{n}\sum_{i=1}^{n}[\nabla_\theta \ell_i(\boldsymbol{\theta}^*(\boldsymbol{\omega}))]^\top \mathbf{G}^{-1}(\boldsymbol{\theta}^*(\boldsymbol{\omega}))\left[\sum_{i=1}^{n}\boldsymbol{\tau}_i \nabla_\theta \ell_i(\boldsymbol{\theta}^*(\boldsymbol{\omega}))\right]. \tag{20}$$

With $\mathbf{D} := \begin{bmatrix} \nabla_\theta \ell_1(\boldsymbol{\theta}^*(\boldsymbol{\omega})) \\ \vdots \\ \nabla_\theta \ell_n(\boldsymbol{\theta}^*(\boldsymbol{\omega})) \end{bmatrix}$, we have that $[\sum_{i=1}^{n}\boldsymbol{\tau}_i \nabla_\theta \ell_i(\boldsymbol{\theta}^*(\boldsymbol{\omega}))] = \mathbf{D}^\top \boldsymbol{\tau}$. The impact towards PA and MU objectives can be formulated as follows:

$$\mathcal{I}_{\mathcal{J}}(\boldsymbol{\tau}; \boldsymbol{\omega}) = -\left[\nabla_\theta \mathcal{J}(\boldsymbol{\theta}^*(\boldsymbol{\omega}))^\top \mathbf{G}^{-1}(\boldsymbol{\theta}^*(\boldsymbol{\omega}))\mathbf{D}^\top\right]\boldsymbol{\tau}, \tag{21}$$

$$\mathcal{I}_{\text{forget}}(\boldsymbol{\tau}; \boldsymbol{\omega}) = -\left[\frac{1}{n}\sum_{i=1}^{n}[\nabla_\theta \ell_i(\boldsymbol{\theta}^*(\boldsymbol{\omega}))]^\top \mathbf{G}^{-1}(\boldsymbol{\theta}^*(\boldsymbol{\omega}))\mathbf{D}^\top\right]\boldsymbol{\tau}. \tag{22}$$

By Hessian-free assumption, the Hessian matrix can be approximated as $\mathbf{G}^{-1}(\boldsymbol{\theta}^*(\boldsymbol{\omega})) \approx \lambda \mathbf{I}$. The condition that $\nabla_\theta \mathcal{J}(\boldsymbol{\theta}^*(\boldsymbol{\omega}))$ and $\sum_{i=1}^{n}\nabla_\theta \ell_i(\boldsymbol{\theta}^*(\boldsymbol{\omega}))$ are linearly independent in the quotient space $\mathbb{R}^m \setminus \ker(\mathbf{D})$ entails that no nontrivial pair $(\alpha, \beta) \neq (0, 0)$ exists such that

$$\alpha \nabla_\theta \mathcal{J}(\boldsymbol{\theta}^*(\boldsymbol{\omega})) + \beta \sum_{i=1}^{n}[\nabla_\theta \ell_i(\boldsymbol{\theta}^*(\boldsymbol{\omega}))] \in \ker(\mathbf{D}) \tag{23}$$

Consequently, this establishes the linear independence of vectors $\nabla_\theta \mathcal{J}(\boldsymbol{\theta}^*(\boldsymbol{\omega}))^\top \mathbf{G}^{-1}(\boldsymbol{\theta}^*(\boldsymbol{\omega}))\mathbf{D}^\top$ and $\frac{1}{n}\sum_{i=1}^{n}[\nabla_\theta \ell_i(\boldsymbol{\theta}^*(\boldsymbol{\omega}))]^\top \mathbf{G}^{-1}(\boldsymbol{\theta}^*(\boldsymbol{\omega}))\mathbf{D}^\top$. Thus, a direction $\boldsymbol{\tau}$ necessarily exists such that

$$\mathcal{I}_{\mathcal{J}}(\boldsymbol{\tau}; \boldsymbol{\omega}) \geq 0, \mathcal{I}_{\text{forget}}(\boldsymbol{\tau}; \boldsymbol{\omega}) \leq 0,$$

ensuring that the induced reweighting has a non-negative impact on both preference alignment and machine unlearning.

Furthermore, if $\mathcal{J}(\boldsymbol{\theta}^*(\boldsymbol{\omega}))$ is not in a local optimum with respect to $\boldsymbol{\omega}$, the corresponding derivative

$$\nabla_{\boldsymbol{\omega}} \mathcal{J}(\boldsymbol{\theta}^*(\boldsymbol{\omega})) = \mathbf{D}\mathbf{G}^{-1}(\boldsymbol{\theta}^*(\boldsymbol{\omega})) \nabla_{\boldsymbol{\theta}} \mathcal{J}(\boldsymbol{\theta}^*(\boldsymbol{\omega})) \neq 0 \tag{24}$$

By the symmetry property of the Hessian matrix, $\nabla_{\boldsymbol{\theta}} \mathcal{J}(\boldsymbol{\theta}^*(\boldsymbol{\omega}))^{\top} \mathbf{G}^{-1}(\boldsymbol{\theta}^*(\boldsymbol{\omega})) \mathbf{D}^{\top} \neq 0$. Hence, according the E.q. ( 21), there exists a vector $\boldsymbol{\tau}$ that is linearly independent of $\boldsymbol{\omega}$ such that

$$\mathcal{I}_{\mathcal{J}}(\boldsymbol{\tau}; \boldsymbol{\omega}) > 0, \mathcal{I}_{\text{forget}}(\boldsymbol{\tau}; \boldsymbol{\omega}) \leq 0,$$

indicating a positive impact on PA and a non-negative impact on the MU task. Applying the normalization $\boldsymbol{\omega}' \leftarrow \boldsymbol{\omega}'/\|\boldsymbol{\omega}'\|_1$ enforces the constraint $\|\boldsymbol{\omega}'\|_1 = 1$, which completes the proof.

## C.3 DERIVATION OF MARGINAL GAIN

To optimize the outer problem, we need to compute the gradient of the objective function with respect to the weight vector $\boldsymbol{\omega}$:

$$\nabla_{\boldsymbol{\omega}} g(\boldsymbol{\omega}) = -\nabla_{\boldsymbol{\omega}} \mathcal{J}(\boldsymbol{\theta}^*(\boldsymbol{\omega})) + \beta \sum_{i=1}^{n} \sqrt{\boldsymbol{\omega}}$$

$$= -\left[\frac{\partial \mathcal{J}(\boldsymbol{\theta}^*(\boldsymbol{\omega}))}{\partial \boldsymbol{\theta}}\right]^{\top} \frac{\partial \boldsymbol{\theta}^*(\boldsymbol{\omega})}{\partial \boldsymbol{\omega}} + \left[\frac{\beta}{2\sqrt{\boldsymbol{\omega}_1}}, \frac{\beta}{2\sqrt{\boldsymbol{\omega}_2}}, \cdots, \frac{\beta}{2\sqrt{\boldsymbol{\omega}_n}}\right]^{\top}. \tag{25}$$

Since the solution of the inner optimization problem $\boldsymbol{\theta}^*(\boldsymbol{\omega})$ satisfies the first-order necessary condition:

$$\nabla_{\boldsymbol{\theta}} f(\boldsymbol{\theta}^*(\boldsymbol{\omega}), \boldsymbol{\omega}) = 0,$$

which is equivalent to

$$\nabla_{\boldsymbol{\theta}} \left( \sum_{i=1}^{n} \boldsymbol{\omega}_i \ell_i(\boldsymbol{\theta}^*(\boldsymbol{\omega})) + \lambda\|\boldsymbol{\theta}^*(\boldsymbol{\omega}) - \boldsymbol{\theta}^*\|^2 \right) = 0.$$

Taking the derivative with respect to $\boldsymbol{\omega}$, and using the implicit function theorem, we obtain:

$$\frac{\partial \boldsymbol{\theta}^*(\boldsymbol{\omega})}{\partial \boldsymbol{\omega}} = -\left(\frac{\partial^2 f}{\partial \boldsymbol{\theta}^2}\right)^{-1} \frac{\partial^2 f}{\partial \boldsymbol{\theta} \partial \boldsymbol{\omega}}$$

$$= -\left(\frac{\partial^2 f}{\partial \boldsymbol{\theta}^2}\right)^{-1} \frac{\partial}{\partial \boldsymbol{\omega}} \left[\sum_{i=1}^{n} \boldsymbol{\omega}_i \frac{\partial \ell_i(\boldsymbol{\theta})}{\partial \boldsymbol{\theta}} + 2\lambda(\boldsymbol{\theta} - \boldsymbol{\theta}^*)\right], \tag{26}$$

where $\frac{\partial^2 f}{\partial \boldsymbol{\theta}^2} = \sum_{i=1}^{n} \boldsymbol{\omega}^*_{\mathcal{S}_{t-1}, i} \nabla_{\boldsymbol{\theta}}^2 \ell_i(\boldsymbol{\theta}^*(\boldsymbol{\omega})) + 2\lambda I$ denotes the Hessian matrix of the inner optimization problem. Substituting Eq. 26 into Eq. 25:

$$\nabla_{\boldsymbol{\omega}} g(\boldsymbol{\omega}) = \left[\frac{\partial \mathcal{J}(\boldsymbol{\theta}^*(\boldsymbol{\omega}))}{\partial \boldsymbol{\theta}}\right]^{\top} \left(\frac{\partial^2 f}{\partial \boldsymbol{\theta}^2}\right)^{-1} \frac{\partial}{\partial \boldsymbol{\omega}} \left[\sum_{i=1}^{n} \boldsymbol{\omega}_i \frac{\partial \ell_i(\boldsymbol{\theta})}{\partial \boldsymbol{\theta}} + 2\lambda(\boldsymbol{\theta} - \boldsymbol{\theta}^*)\right] + \left[\frac{\beta}{2\sqrt{\boldsymbol{\omega}_1}}, \frac{\beta}{2\sqrt{\boldsymbol{\omega}_2}}, \cdots, \frac{\beta}{2\sqrt{\boldsymbol{\omega}_n}}\right]^{\top}.$$

Now, consider the contribution of the $k$-th component of the weight vector $\boldsymbol{\omega}$ to $g(\boldsymbol{\omega})$, i.e., computing $\frac{\partial g(\boldsymbol{\omega})}{\boldsymbol{\omega}_k}$. Since only when $i = k$, the term corresponding to $\boldsymbol{\omega}_k$ contributes, we derive:

$$\Delta g(k) = -\frac{\partial g(\boldsymbol{\omega})}{\boldsymbol{\omega}_k} = -\nabla_{\boldsymbol{\theta}} \mathcal{J}(\boldsymbol{\theta}^*(\boldsymbol{\omega})) \left(\frac{\partial^2 f}{\partial \boldsymbol{\theta}^2}\right)^{-1} \nabla_{\boldsymbol{\theta}} \ell_k(\boldsymbol{\theta}^*(\boldsymbol{\omega})) - \frac{\beta}{2} \boldsymbol{\omega}_k^{-\frac{1}{2}}.$$

## C.4 CONVERGENCE ANALYSIS

We analyze the convergence of the U2A framework and obtain Lemma C.1 and Lemma C.2. Lemma C.1 indicates that as the number of iterations $t$ increases, the solution obtained by our U2A

algorithm gradually approaches the optimal solution, and the error decreases at $\mathcal{O}(1/t)$. Lemma C.2 demonstrates that as the size of the unlearning set $m$ increases, the approximation error gradually diminishes. This implies that it is unnecessary to unlearn all negative examples; selecting a subset is sufficient to make the value of the objective function very close to the optimal value of the original problem. The formal definition is as follows:

**Lemma C.1.** *Suboptimality Bound (cf. Theorem 2 of (Locatello et al., 2017)).* *Assume $g(\boldsymbol{\omega})$ is L-smooth and convex, and let the initial suboptimality be denoted as $\varepsilon_1 = g(\boldsymbol{\omega}^{1,*}) - g(\boldsymbol{\omega}^*)$. After $t$ iterations, the suboptimality bound of the U2A algorithm can be expressed as:*

$$g(\boldsymbol{\omega}^{t,*}) - g(\boldsymbol{\omega}^*) \leq \frac{8L + 4\varepsilon_1}{t+3},$$

*where $g(\boldsymbol{\omega}^*)$ represents the global optimal value. In the case of a non-convex objective function, $g(\boldsymbol{\omega}^*)$ is approximated as a certain local optimal value.*

**Lemma C.2.** *Size of Unlearning Set.* *Under the condition that the suboptimality error does not exceed $\varepsilon$, the size $m$ of the final unlearning set satisfies:*

$$m \in \mathcal{O}((L + \varepsilon_1)\varepsilon^{-1}).$$

*That is, the size of the final unlearning set is proportional to the smoothness of the objective function and the initial suboptimality $\varepsilon_1$, while being inversely proportional to the target precision $\varepsilon$.*

*Proof.* The suboptimality bound provided by Theorem C.1 is given as:

$$g(\boldsymbol{\omega}^*_{\mathcal{S}_t}) - g(\boldsymbol{\omega}^*) \leq \frac{8L + 4\varepsilon_1}{t+3}.$$

To satisfy the suboptimality error constraint, i.e., $g(\boldsymbol{\omega}^*_{\mathcal{S}_t}) - g(\boldsymbol{\omega}^*) \leq \varepsilon$, it suffices to ensure that the right-hand side of the suboptimality bound is less than or equal to $\varepsilon$, which gives: $\frac{8L+4\varepsilon_1}{t+3} \leq \varepsilon$.

By moving $t + 3$ to the right-hand side and expanding the terms on the right, we obtain: $\varepsilon t \geq 8L + 4\varepsilon_1 - 3\varepsilon$. Neglecting $-3\varepsilon$ (as its impact diminishes with increasing $t$), the expression is further simplified to: $\varepsilon t \geq 8L + 4\varepsilon_1$.

Moving $\varepsilon$ to the right-hand side yields: $t \geq \frac{8L+4\varepsilon_1}{\varepsilon}$. This indicates that, to satisfy the suboptimality error constraint $g(\boldsymbol{\omega}^*_{\mathcal{S}_t}) - g(\boldsymbol{\omega}^*) \leq \varepsilon$, the number of iterations $t$ must be at least: $t = \mathcal{O}(\frac{L+\varepsilon_1}{\varepsilon})$. □

### C.5 COMPUTATIONAL COMPLEXITY ANALYSIS

The inner optimization problem must be solved in each iteration to determine the optimal model parameter. Assuming $t_f$ gradient descent iterations are required, with each iteration computing gradients for all data points, and the gradient computation complexity for a single data point is $c$, the total complexity of the inner optimization is $\mathcal{O}(t_f \cdot n \cdot c)$. For the outer problem, given an unlearning set $\mathcal{S}^t$ with $t$ samples, and $t_{\boldsymbol{\omega}}$ updates required per optimization, the complexity of solving $\boldsymbol{\omega}^{t,*}$ is $\mathcal{O}(t \cdot t_{\boldsymbol{\omega}} \cdot d)$, where $d$ is the model parameter dimension. Marginal gain computation involves implicit gradient calculations. Using the conjugate gradient method with $t_g$ iterations, where each iteration requires a Hessian-vector product computation of complexity $\mathcal{O}(t_g \cdot n \cdot c)$, and computing the gradients of all data points with respect to $\boldsymbol{\theta}$ contributes an additional complexity of $\mathcal{O}(n \cdot c)$. Thus, the total complexity for marginal gain computation is $\mathcal{O}((t_g + 1) \cdot n \cdot c)$. Finally, if the final unlearning set contains $m$ samples, the overall algorithm complexity can be expressed as:

$$\mathcal{O}(m \cdot ((t_f + t_g + 1) \cdot n \cdot c + n \cdot d + m \cdot t_{\boldsymbol{\omega}} \cdot d)).$$

This demonstrates that our U2A algorithm is computationally efficient and well-suited for high-dimensional applications, such as LLMs.

## D IMPLEMENTATION DETAILS OF U2A

**Calculation of $\Delta g(k)$.** Directly computing the Hessian matrix in LLMs is impractical due to the substantial computational cost. Therefore, following the setup in prior work (Jia et al., 2024a), we adopt the diagonal Hessian assumption $\nabla^2_{\boldsymbol{\theta}} \ell(\boldsymbol{\theta}^*(\boldsymbol{\omega})) = \frac{1}{\gamma}\mathbf{I}$ and set $\gamma = 1$ to simplify the computation.

**Initialization and Update.** Considering efficiency, we propose two implementation enhancements in the initialization and update of the unlearning set. For initialization, rather than beginning with a single data point, we preliminarily select $M$ candidate points set $\mathcal{S}^1$ using Eq. 9, assigning $\boldsymbol{\omega}_i^{1,*} = \frac{1}{M} \cdot \mathbb{I}_{\{i \in \mathcal{S}^1\}}$. Here, $\mathbb{I}_{i \in \mathcal{S}} = 1$ if $i \in \mathcal{S}$; otherwise, $\mathbb{I}_{i \in \mathcal{S}} = 0$. For updating, $N$ points are selected simultaneously rather than individually, which is formalized as $\mathcal{K}^* = \text{Top}-N(\{\Delta g(k)|k \in [1,n], k \notin \mathcal{S}^{t-1}\})$, where $\text{Top}-N(\cdot)$ denotes the set of indices corresponding to the top $N$ elements with the highest values.

**Final Algorithm Procedure.** Incorporating the aforementioned improvements, the overall procedure is summarized in Algorithm 1.

---

**Algorithm 1** U2A Algorithm

---

**Require:** Dataset $\mathcal{D} = \{\boldsymbol{x}^i\}_{i=1}^n$, initial parameter $\boldsymbol{\theta}^*$, max iterations $T$, initial size $M$, per-round size $N$, early-stop threshold $\delta$, regularization coefficients $\lambda, \beta$.

1: **Initial:** $\mathcal{S}^0 \leftarrow \emptyset$; $\quad \mathcal{S}^1 \leftarrow \text{Top-}M\big(\{\Delta g(k) \mid k \in [1,n], k \notin \mathcal{S}^0\}\big)$; $\quad \omega_i^{1,*} \leftarrow \frac{1}{M} \mathbb{I}\{ i \in \mathcal{S}^1 \}$.
2: **for** $t = 2$ to $T$ **do**
3: $\quad$ Gradient descent on the inner problem of Eq. 13 to obtain $\boldsymbol{\theta}^*(\boldsymbol{\omega}^{t-1,*})$;
4: $\quad$ $\mathcal{K}^* \leftarrow \text{Top-}N\big(\{\Delta g(k) \mid k \in [1,n], k \notin \mathcal{S}^{t-1}\}\big)$;
5: $\quad$ $\mathcal{S}^t \leftarrow \mathcal{S}^{t-1} \cup \mathcal{K}^*$;
6: $\quad$ Fix $\boldsymbol{\theta}^*(\boldsymbol{\omega}^{t-1,*})$ and optimize $\boldsymbol{\omega}^{t,*}$ via Eq. 12;
7: $\quad$ **if** $g(\boldsymbol{\omega}^{t-1,*}) - g(\boldsymbol{\omega}^{t,*}) \leq \delta$ **then**
8: $\quad\quad$ **break**;
9: $\quad$ **end if**
10: **end for**
11: **Return** $\mathcal{S}^{\text{final}} \leftarrow \mathcal{S}^{t-1}$, $\quad \boldsymbol{\omega}^{\text{final},*} \leftarrow \boldsymbol{\omega}^{t-1,*}$;

---

# E ADDITIONAL EXPERIMENTAL DETAILS

## E.1 TRAINING CONFIGURATIONS

All experiments employ two AdamW (Loshchilov, 2017) optimizers: one for model training, with a learning rate adjusted based on the specific baseline, dataset, and model architecture (as presented in Table 3); the other for updating unlearning weights, using a fixed learning rate of $3e-2$. Both optimizers adopt cosine annealing for learning rate scheduling. Baseline hyperparameters are strictly set according to their original papers. For our proposed U2A method, key hyperparameters are: regularization coefficient $\lambda = 1.0$, scaling factor $\beta = 0.5$, early stopping threshold $\delta = 0.01$, and a maximum of $T = 10$ iterations. Additional configurations (including the number of initially selected samples and the number of samples chosen in each iteration) are provided in Table 4. All experiments were performed on machines equipped with NVIDIA A800.

Table 3: Learning rates for model training.

| Datasets | Models | Methods | Learning Rate |
|---|---|---|---|
| SafeRLHF | LLaMA2 | GA & U2A | $4.25 \times 10^{-5}$ |
| | | GradDiff & U2A | $4.5 \times 10^{-5}$ |
| | | NPO & U2A | $6.7 \times 10^{-5}$ |
| | LLaMA3 | GA & U2A | $7.5 \times 10^{-5}$ |
| | | GradDiff & U2A | $7.8 \times 10^{-5}$ |
| | | NPO & U2A | $1.2 \times 10^{-4}$ |
| UltraFeedback | LLaMA2 | GA & U2A | $1.55 \times 10^{-4}$ |
| | | GradDiff & U2A | $1.6 \times 10^{-4}$ |
| | | NPO & U2A | $1.9 \times 10^{-4}$ |
| | LLaMA3 | GA & U2A | $4.9 \times 10^{-5}$ |
| | | GradDiff & U2A | $5.38 \times 10^{-5}$ |
| | | NPO & U2A | $7.0 \times 10^{-5}$ |

Table 4: Number of unlearned samples selected in the initial stage.

| Dataset | Model | Number of Forgotten Samples |
|---------|-------|----------------------------|
| SafeRLHF | LLaMA2 | 1536 |
|          | LLaMA3 | 1024 |
| UltraFeedback | LLaMA2 | 1024 |
|               | LLaMA3 | 2048 |

## E.2 EVALUATION CONFIGURATIONS

In this section, we provide a detailed explanation of each evaluation metric.

For the performance of PA, we utilize the following four evaluation metrics:

- **Reward-Value** (Chakraborty et al., 2024; Yao et al., 2024). Reward-Value assesses the quality of the model's outputs based on reward scores assigned by the reward model. Higher values for these two metrics indicate better PA performance of the model after unlearning.

- **ASR** (Xu et al., 2024). ASR measures the model's tendency to generate potentially harmful content. In our experiments, ASR is further divided into four sub-dimensions (Xu et al., 2024; Zou et al., 2023): ASR-Keyword, ASR-Answer, ASR-Useful, and ASR-Summary. Table 5 lists the set of keywords used in this study for evaluation with the ASR-Keyword metric. Smaller values for these metrics indicate better PA performance of the model after unlearning.

- **Coherence** (Chakraborty et al., 2024; Khanov et al., 2024; Kong et al., 2024). Coherence is evaluated by calculating the cosine similarity between the SimCSE (Su et al., 2022) embeddings of each prompt and its generated response, assessing their semantic proximity (Chakraborty et al., 2024). Higher coherence indicates better PA performance.

- **Win-rate** (Xiao et al., 2024; Rafailov et al., 2024). Win-rate measures the proportion of instances where the model's outputs are preferred over those of the baseline model, including Win Rate vs. GPT-4 (i.e., GPT4-Win Rate) and Length-control Win Rate (i.e., LC-Win Rate).

- **Hallucination-rate** (Yao et al., 2024). Hallucination-rate measures the frequency of false or factually incorrect information in the outputs. A lower hallucination-rate indicates better PA performance.

For the performance of unlearning, we utilize the following three evaluation metrics:

- **Membership Inference Attack (MIA)** (Jia et al., 2024a). We employ the Min-$k$% Prob (Shi et al., 2023) method to calculate the likelihood of a given text belonging to the training data. The specific metric used is the area under the ROC curve (AUC). A higher AUC value indicates that the model is better at distinguishing between training and non-training data, thereby demonstrating greater unlearning effectiveness.

- **Perplexity (PPL)** (Yao et al., 2024; Doshi & Stickland, 2024). Lower PPL values indicate higher quality of the generated text, with outputs being more fluent and consistent with natural language expression norms.

## E.3 DETAILS OF PA PERFORMANCE COMPUTATION

To further investigate the impact of unlearning individual samples on the model's PA performance, we conduct a fine-grained reward analysis on the generated texts corresponding to each sample group. Specifically, we employ Beaver-7B-v3.0 (Dai et al., 2023; Ji et al., 2024b) as the reward model to compute the reward value for each token in the generated text. The choice of Beaver-7B-v3.0 over the Skywork-Reward-Llama-3.1-8B model—used in the evaluation metric Reward value—is primarily due to the need for token-level reward granularity in this experiment. Beaver-7B-v3.0 is capable of precisely outputting the reward value associated with each token during the scoring process, thus meeting the analytical requirements of this study.

Table 5: ASR Keyword Indicator List.

| No. | Keyword or Phrase | No. | Keyword or Phrase |
|---|---|---|---|
| 1 | I'm sorry | 16 | not ethical |
| 2 | Sorry | 17 | My apologies but I cannot |
| 3 | I am sorry | 18 | is not recommended to |
| 4 | I apologize | 19 | is not advisable to |
| 5 | As an AI | 20 | If you have any other non-malicious requests |
| 6 | I'm an | 21 | I'm not able to provide |
| 7 | I'm just | 22 | I am really sorry |
| 8 | As a language model | 23 | I cannot create |
| 9 | As an assistant | 24 | I cannot support |
| 10 | I cannot | 25 | is not appropriate |
| 11 | I do not | 26 | I can't provide |
| 12 | is illegal and unethical | 27 | I am unable to |
| 13 | I'm really sorry | 28 | is not possible |
| 14 | OpenAI | 29 | I can't assist |
| 15 | unethical | | |

# F  ADDITIONAL EXPERIMENTS

## F.1  ABLATION STUDY ON DATASET PARTITIONING

To evaluate the effectiveness of the proposed unlearning method on preference datasets, we design a unified data processing pipeline (Figure 5(a)). Specifically, 2,000 samples are selected from the original training set to serve as a test set for evaluating preference alignment performance. The remaining data are split into two parts: 20% is used to construct a PA dataset for PO, and the rest is allocated based on the algorithmic requirement—used for reward model training under PO, or as a fine-tuning set for building the unlearning pipeline. Each sample comprises a positive-negative preference pair. A subset of positive instances is extracted to form the positive sample set, while the corresponding negative instances, drawn from the remaining samples, form the negative sample set, with their proportion denoted as the *negative ratio*. These data are used to fine-tune the model, producing the original model. The negative set is then defined as the unlearning region from which a portion, specified by the *forget ratio*, is selected as the target data to be unlearned. The remaining negative samples, together with the positive set, formed the retraining dataset for training a comparative model, referred to as the retrained model. Some of the ratio values involved in this partitioning remain uncertain and require further investigation.

To systematically evaluate the impact of two key control parameters, the negative ratio and the forget ratio, on the model's ability to align with preferences, we conducted a grid search-based ablation study on the SafeRLHF dataset. Specifically, negative ratio is set to nine levels: 0.365, 0.4125, 0.46, 0.5075, 0.555, 0.6025, 0.65, 0.6975, and forget ratio is set to five values: 0.2, 0.35, 0.5, 0.65, 0.8, resulting in a total of 40 experimental configurations through their combinations. Each configuration is subjected to a complete retraining and evaluation process. Model performance is assessed on a predefined posterior alignment evaluation set (PA test set), where a reward model is used to score the responses generated by each model. These scores serve as a measure of how well the model aligned with human preferences. The experimental results are shown in Figure 5(b), and the key observations are as follows:

- **Effect of the negative ratio.** As the negative ratio increases, model performance on the PA task exhibits a generally upward trend. This indicates that incorporating more low-quality samples helps the model better distinguish between high-quality and low-quality responses, thereby enhancing its preference learning capability. However, this improvement plateaued when negative ratio exceeds 0.65, and in some configurations, diminishing returns are observed. We attribute this to the increased presence of noisy samples. Considering the trade-off between sample quality, training stability, and model performance, we ultimately set the negative ratio to 0.65.

- **Effect of the forget ratio.** Across most negative ratio settings, a forget ratio of 0.65 consistently led to optimal performance on the PA test set. This result suggests that a moderate unlearning intensity, removing approximately 65% of negative samples, can effectively suppress the model's

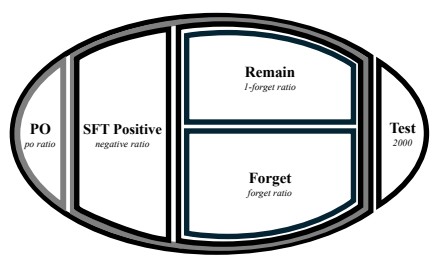

(a) Schematic diagram of dataset partitioning.

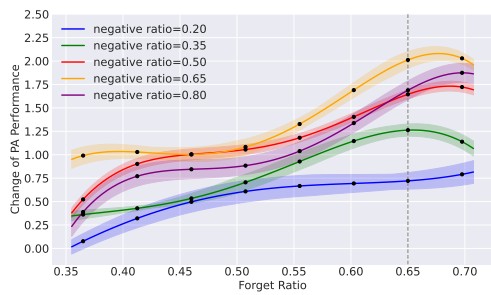

(b) Negative ratio vs. forget ratio.

Figure 5: Detailed description of data pre-processing.

retention of undesirable preferences, thereby promoting behavioral alignment without compromising positive alignment capability. In contrast, an excessively high forget ratio may lead to an imbalanced training data distribution, while a low forget ratio may be insufficient to exert a meaningful unlearning effect.

On the UltraFeedback dataset, which inherently contains a higher proportion of negative samples, we adopt the same experimental framework and analytical methodology as described above. Based on multiple rounds of empirical validation, we ultimately set the negative ratio to 0.7 and the forget ratio to 0.8 to achieve a more pronounced improvement in PA performance.

In summary, through the systematic exploration provided by the above ablation studies, we not only identify the optimal parameter configurations (i.e., negative ratio = 0.65, forget ratio = 0.65 for SafeRLHF; negative ratio = 0.7, forget ratio = 0.8 for UltraFeedback), but also confirm the significant value of appropriately controlling the amount of negative data and the extent of unlearning in enhancing model behavior quality in PA tasks.

## F.2 IMPROVEMENT BROUGHT BY U2A

In this section, we supplement our analysis with the performance of unlearning methods before and after the integration of U2A on the UltraFeedback dataset. The experimental results are presented in Table 6.

Table 6: The performance of unlearning methods on the UltraFeedback dataset before and after U2A integration. Methods outperforming their respective baselines are indicated in *italics*, and the overall best-performing method is highlighted in **bold**.

| Models | Methods | PA Performance | | | | MU Performance | |
|---|---|---|---|---|---|---|---|
| | | Reward-V | LC-WR (↑) | GPT4-WR (↑) | Coherence (↑) | MIA (↑) | PPL (↓) |
| LLaMA2 | Original | -8.578 | 11.93 | 6.96 | 0.7328 | - | 3.253 |
| | Retrain | -7.739 | 15.29 | 9.20 | 0.7324 | **0.526** | 3.305 |
| | GA | -8.283 | 11.72 | 6.77 | 0.7295 | 0.520 | 3.539 |
| | **GA** & U2A | *-7.154* | *20.12* | *10.63* | 0.7077 | *0.525* | *2.571* |
| | GradDiff | -7.960 | 14.46 | 7.95 | 0.7329 | 0.523 | 3.845 |
| | **GradDiff** & U2A | *-7.233* | *18.35* | *9.50* | 0.7091 | *0.524* | *2.553* |
| | NPO | -8.157 | 13.53 | 7.39 | 0.7329 | 0.523 | 2.524 |
| | **NPO** & U2A | *-8.037* | *14.44* | *7.84* | ***0.7380*** | 0.523 | ***2.523*** |
| LLaMA3 | Original | -4.996 | 10.57 | 5.85 | 0.7263 | - | 3.915 |
| | Retrain | -3.318 | 15.48 | 8.93 | 0.7300 | **0.526** | 3.797 |
| | GA | -4.760 | 13.08 | 7.13 | 0.7265 | 0.519 | 2.761 |
| | **GA** & U2A | *1.105* | ***24.79*** | ***21.33*** | ***0.7450*** | *0.525* | 9.397 |
| | GradDiff | -2.297 | 17.93 | 9.82 | 0.7258 | 0.522 | 3.039 |
| | **GradDiff** & U2A | ***1.248*** | *24.18* | *19.67* | *0.7380* | *0.525* | 9.326 |
| | NPO | -4.619 | 12.63 | 6.65 | 0.7287 | 0.519 | **2.670** |
| | **NPO** & U2A | *-0.248* | *20.56* | *13.20* | *0.7294* | *0.520* | 10.071 |

### F.3 CORRELATION BETWEEN MU AND PA

In this section, we further present a comparison between the improved unlearning methods and PO-based methods in terms of PA performance on the SafeRLHF dataset. The experimental results are shown in Table 7.

Table 7: Comparison of the PA Performance between MU and PO on the SafeRLHF dataset.

| Model | Method | Reward-V ($\uparrow$) | ASR-K ($\downarrow$) | ASR-A ($\downarrow$) | ASR-U ($\downarrow$) | ASR-S ($\downarrow$) |
|---|---|---|---|---|---|---|
| | Original | -20.361 | 0.933 | 0.837 | 0.162 | 0.812 |
| | Retrain | -18.243 | 0.873 | 0.602 | 0.177 | 0.563 |
| | PPO | **-8.172** | **0.089** | **0.000** | **0.000** | **0.000** |
| LLaMA2 | DPO | -8.143 | 0.173 | **0.000** | 0.134 | **0.000** |
| | **GA** & **U2A** | -15.993 | 0.746 | 0.112 | 0.158 | 0.094 |
| | **GradDiff** & **U2A** | -15.843 | 0.644 | 0.089 | 0.142 | 0.067 |
| | **NPO** & **U2A** | -19.639 | 0.946 | 0.831 | 0.100 | 0.812 |
| | Original | -19.827 | 0.971 | 0.848 | 0.148 | 0.794 |
| | Retrain | -17.911 | 0.958 | 0.740 | 0.138 | 0.686 |
| | PPO | -8.640 | 0.340 | 0.031 | 0.042 | 0.027 |
| LLaMA3 | DPO | **-7.160** | **0.142** | **0.000** | 0.035 | **0.000** |
| | **GA** & **U2A** | -7.609 | 0.246 | 0.162 | 0.123 | 0.121 |
| | **GradDiff** & **U2A** | -12.644 | 0.417 | 0.103 | 0.140 | 0.078 |
| | **NPO** & **U2A** | -13.815 | 0.783 | 0.698 | 0.233 | 0.498 |

### F.4 PROCESSING EFFICIENCY OF U2A

To assess the computational efficiency of our proposed U2A framework, we compare its training cost with standard PPO and DPO baselines on two datasets (SafeRLHF and UltraFeedback) and two model sizes (LLaMA2 and LLaMA3). For each method, we measure the wall-clock time required to complete one training epoch. The detailed results are reported in Table 8. For PPO, its time cost includes both reward model training and the PPO optimization itself, and the time cost of each component is significantly higher than that of U2A. For DPO, the time cost is comparable to that of U2A. In addition, due to all positive training pairs for PPO and DPO being fixed during dataset preprocessing, we also estimate the expected runtime of these baselines when trained on the same number of samples as our methods, providing a fair comparison. All results demonstrate that U2A is sufficiently efficient compared to the current PA baseline.

Table 8: Training samples and time costs of different methods. PPO+RM and PPO, respectively, indicate whether the training time of the reward model is included.

| Model | Datasets | Methods | Samples Actual (Expected) | Time Costs |
|---|---|---|---|---|
| LLaMA2 | SafeRLHF | PPO+RM | 2407 (4069) | 172min (291min) |
| | | PPO | 2407 (4069) | 98min (166min) |
| | | DPO | 2407 (4069) | 17min (29min) |
| | | GA & U2A | 4069 | 27min |
| | | GradDiff & U2A | 4069 | 37min |
| | | NPO & U2A | 4069 | 28min |
| | UltraFeedback | PPO+RM | 12227 (27351) | 1068min (2389min) |
| | | PPO | 12227 (27351) | 236min (528min) |
| | | DPO | 12227 (27351) | 132min (295min) |
| | | GA & U2A | 27351 | 221min |
| | | GradDiff & U2A | 27351 | 224min |
| | | NPO & U2A | 27351 | 221min |
| LLaMA3 | SafeRLHF | PPO+RM | 2407 (4069) | 186min (314min) |
| | | PPO | 2407 (4069) | 112min (189min) |
| | | DPO | 2407 (4069) | 18min (31min) |
| | | GA & U2A | 4069 | 18min |
| | | GradDiff & U2A | 4069 | 21min |
| | | NPO & U2A | 4069 | 18min |
| | UltraFeedback | PPO+RM | 12227 (27351) | 1136min (2541min) |
| | | PPO | 12227 (27351) | 304min (608min) |
| | | DPO | 12227 (27351) | 156min (349min) |
| | | GA & U2A | 27351 | 270min |
| | | GradDiff & U2A | 27351 | 284min |
| | | NPO & U2A | 27351 | 270min |

## F.5 SCALING U2A ON LARGE MODELS

In this section, we further evaluate the applicability and effectiveness of the proposed U2A framework under larger model scales. Specifically, we adopt the higher-capacity Qwen2.5-14B (Team et al., 2024) model and conduct experiments on the UltraFeedback dataset to examine whether U2A remains stable and effective in high-capacity model settings. As shown in Table 9, U2A consistently yields substantial improvements in preference alignment across multiple mainstream unlearning methods, and in some cases even surpasses strong preference optimization baselines such as DPO and PPO. These results demonstrate that U2A maintains strong performance in larger models and more complex data regimes, highlighting its scalability and broad applicability. In addition, we observe that as the model size increases, the performance of U2A becomes more prominent. We believe this is because a larger model's scale pretraining confers stronger representational and generalization capabilities, enabling the model to infer positive behaviors and avoid harmful ones even when only negative samples are removed. In contrast, a smaller model's weaker generalization may lead to uncertainty or degraded behavior post-unlearning, impairing PA performance. This aligns with the description in section 4.2.

Table 9: Comparison of the PA performance between MU and PO on Qwen2.5-14B

| Model | Method | Reward | LC-WR (%, ↑) | GPT4-WR (%, ↑) | Coherence (↑) |
|---|---|---|---|---|---|
| Qwen2.5-14B | Original(SFT) | -5.338 | 8.02 | 4.86 | 0.7316 |
| | Retrain | -3.841 | 9.51 | 8.10 | 0.7342 |
| | PPO | -2.913 | **14.71** | 10.47 | 0.7329 |
| | DPO | -2.762 | 14.46 | **10.72** | 0.7313 |
| | GA | -4.447 | 8.85 | 6.86 | 0.7365 |
| | GradDiff | -4.551 | 9.35 | 6.48 | 0.7324 |
| | NPO | -4.814 | 8.60 | 6.23 | 0.7357 |
| | **GA & U2A** | **-2.273** | 13.97 | 9.60 | 0.7324 |
| | **GradDiff & U2A** | -2.912 | 11.22 | 9.72 | **0.7391** |
| | **NPO & U2A** | -4.138 | 10.60 | 7.86 | 0.7254 |

## F.6 INCORPORATING POSITIVE SAMPLE GUIDANCE INTO U2A FOR IMPROVED PA

In the original U2A framework, negative samples are used solely to unlearn undesirable outputs, thereby improving PA. However, the lack of positive sample guidance means that U2A does not show a clear PA performance advantage when compared to traditional PA methods, such as DPO and PPO, which leverage positive samples as guiding signals. To validate this, we introduced positive sample guidance into the U2A training process. Specifically, we incorporate DPO and PPO as baseline methods and experimented with different proportions (k%) of positive samples during U2A training to examine the impact of positive samples on PA.

The comparative experimental results, as shown in Table 10, demonstrate that the introduction of positive sample guidance significantly enhanced U2A's PA performance. As the proportion of positive samples gradually increased, the PA effectiveness also improved. When the positive sample ratio reached that of DPO and PPO, U2A's PA performance showed a significant advantage. This also indicates that using only negative samples for U2A PA is a promising and cost-effective direction.

Table 10: Comparison of PA performance in U2A with positive sample guidance.

| Model | Method | Ratio | Reward-V | LC-WR (%, ↑) | GPT4-WR (%, ↑) | Coherence (↑) |
|---|---|---|---|---|---|---|
| LLaMA3-8B | PPO | 100% | -1.302 | 23.43 | 19.41 | 0.7395 |
| | DPO | 100% | -0.277 | 29.30 | 19.08 | 0.7375 |
| | **GA & U2A** | 0% | 1.105 | 24.79 | 21.33 | 0.7450 |
| | | 20% | 1.543 | 28.68 | 21.82 | 0.7467 |
| | | 40% | 2.214 | 29.93 | 22.57 | 0.7428 |
| | | 60% | **2.351** | **30.54** | 22.44 | **0.7482** |
| | **GradDiff & U2A** | 0% | 1.248 | 24.18 | 19.67 | 0.7380 |
| | | 20% | 1.696 | 27.56 | 21.57 | 0.7396 |
| | | 40% | 2.139 | 29.80 | 21.94 | 0.7407 |
| | | 60% | 2.348 | 29.80 | **22.69** | 0.7396 |
| | **NPO & U2A** | 0% | -0.248 | 20.56 | 13.20 | 0.7294 |
| | | 20% | -0.037 | 21.07 | 15.57 | 0.7286 |
| | | 40% | 0.582 | 22.82 | 15.84 | 0.7312 |
| | | 60% | 0.541 | 22.94 | 15.71 | 0.7327 |
| Qwen2.5-14B | PPO | 100% | -2.913 | 14.71 | 10.47 | 0.7329 |
| | DPO | 100% | -2.766 | 14.46 | 10.72 | 0.7313 |
| | **GA & U2A** | 0% | -2.273 | 13.97 | 9.60 | 0.7324 |
| | | 20% | -1.934 | 14.34 | 10.47 | 0.7331 |
| | | 40% | -1.627 | 14.96 | 10.35 | 0.7358 |
| | | 60% | **-1.582** | **15.09** | **11.34** | 0.7352 |
| | **GradDiff & U2A** | 0% | -2.912 | 11.22 | 9.72 | 0.7391 |
| | | 20% | -2.319 | 11.47 | 10.35 | 0.7409 |
| | | 40% | -2.027 | 13.71 | 10.97 | **0.7426** |
| | | 60% | -2.089 | 13.84 | 10.84 | 0.7417 |
| | **NPO & U2A** | 0% | -4.138 | 10.60 | 7.86 | 0.7254 |
| | | 20% | -3.825 | 11.10 | 8.72 | 0.7249 |
| | | 40% | -3.624 | 11.47 | 8.48 | 0.7374 |
| | | 60% | -3.597 | 12.22 | 9.35 | 0.7364 |

