# OpenReview forum: "Bridging the Gap Between Preference Alignment and Machine Unlearning"
_ICLR.cc/2026/Conference — ICLR 2026 Conference Withdrawn Submission_

### Official Review · Reviewer_MF9N · 2025-10-25

**Soundness:** 3
**Presentation:** 3
**Contribution:** 3
**Rating:** 6
**Confidence:** 2

**Summary:**

The paper proposes a principled connection between Preference Alignment (PA) and Machine Unlearning (MU) for LLMs. Using a bi-level formulation, the authors quantify how unlearning a specific negative example affects PA and then build U2A (Unlearning-to-Align), which selects and reweights forget samples to maximize PA under a sparsity constraint.

**Strengths:**

A principled bridge between MU and PA with a clean sensitivity view (negative vs positive cosine) and an actionable marginal-gain selector; mirror-descent on simplex gives closed-form weight updates and encourages sparse, high-impact forgetting.
Clear theoretical intuition for when/why unlearning helps alignment; the cosine-of-gradients view is compelling and actionable.
Bi-level design is reasonable and supported by convergence & complexity analysis.
Good ablations.

**Weaknesses:**

The theory uses Hessian PSD & diagonal approximations; stress-testing against curvature mis-specification would be valuable.

Results are on three popular PA datasets; more diversity (such as safety red-teaming, multilingual, domain tasks) would help generality.

While U2A is competitive in some settings, it trails DPO/PPO in others (e.g., with weaker base models / noisier negatives). A controlled compute-matched study would clarify trade-offs.

**Questions:**

see Weaknesses

---

> ### Author Response · Authors · 2025-11-26
>
> >**Weakness 1:** The theory uses Hessian PSD & diagonal approximations; stress-testing against curvature mis-specification would be valuable.
>
> **Answer:** Thank you for your valuable comments. We agree that stress-testing the diagonal approximation of the Hessian is indeed important in traditional machine learning research. However, in the context of LLMs, the direct computation of the full Hessian is practically infeasible due to stringent limitations on computational resources and memory. This also implies that stress-testing Hessian approximation methods for LLMs cannot rely on an exact gold-standard benchmark. Given this practical constraint, existing literature [1,2,3] commonly adopts a scaled identity matrix as a surrogate for the full Hessian in approximation and analysis, and our work follows this established practice. We hope this explanation helps clarify our design choices, and we sincerely appreciate your constructive feedback.
>
>
> >**Weakness 2:** Results are on three popular PA datasets; more diversity (such as safety red-teaming, multilingual, domain tasks) would help generality.
>
> **Answer:** Thank you for your valuable feedback and constructive suggestions. We greatly appreciate your recommendation to introduce more diverse datasets into the experiments, such as security red teams, multilingual data, and domain-specific tasks. As you rightly pointed out, this would indeed help improve the generalization capability of our framework.
>
> However, we chose to focus on these three popular preference alignment datasets because they are widely used in the literature and allow for effective comparison with existing alignment methods. We recognize that for other tasks, relevant comparative baseline methods—such as preference alignment methods—might not be easily applicable, which is why we did not consider them in this experiment. On the other hand, this aligns with the core objective of our work, which is to demonstrate the effectiveness of the U2A framework on these commonly used PA datasets and validate that U2A, when using only negative samples, can achieve alignment results comparable to existing methods that use both positive and negative samples. With this result, we can further highlight that using negative samples for U2A preference alignment is not only a promising direction but also an effective and low-cost method. Once again, thank you for your thorough review and valuable insights into our work.
>
>
> >**Weakness 3:** While U2A is competitive in some settings, it trails DPO/PPO in others (e.g., with weaker base models / noisier negatives). A controlled compute-matched study would clarify trade-offs.
>
> **Answer:** The unlearning method based on the U2A improvement can achieve comparable results to traditional PA methods, such as PPO and DPO. The reason it does not significantly outperform PPO and DPO is that the latter methods use additional positive examples as guiding signals, but the high cost of positive examples led to their exclusion in our setup.
>
> To further validate U2A's practicality, we conduct an additional experiment that introduces positive examples into the U2A framework. More specifically, we incorporate various proportions of positive examples (k%) during the U2A training process. The comparison results with PPO and DPO, as shown in Table 10 in Appendix F.6, clearly demonstrate that the performance of U2A improves continuously as the number of positive examples increases. Moreover, under the same number of positive examples, U2A consistently outperforms both PPO and DPO, indicating that introducing positive sample guidance significantly enhances preference alignment.
>
> Additionally, as model size increases, the performance of U2A also strengthens, as shown in Tables 9 and 10. We believe this is because a larger model's scale pretraining confers stronger representational and generalization capabilities, enabling the model to infer positive behaviors and avoid harmful ones even when only negative samples are removed. In contrast, a smaller model's weaker generalization may lead to uncertainty or degraded behaviors after unlearning, impairing preference alignment performance. This aligns with the description we provided in lines 409-417 of the manuscript, suggesting that using only negative samples for U2A to perform preference alignment is a promising and cost-effective direction.
>
> We have included this new analysis in Appendix F.6.
>
>
> #### **Reference**
>
> [1] Zhang, Yihua, et al. "Revisiting and advancing fast adversarial training through the lens of bi-level optimization." ICML 2022.
>
> [2] Fan, Chongyu, et al. "SalUn: Empowering Machine Unlearning via Gradient-based Weight Saliency in Both Image Classification and Generation." ICLR 2023.
>
> [3] Jia, Jinghan, et al. "Wagle: Strategic weight attribution for effective and modular unlearning in large language models." NeurIPS 2024.

---

> > ### Author Response · Authors · 2025-11-28
> > **We welcome any remaining questions**
> >
> > Dear Reviewer MF9N,
> >
> > Thank you for your valuable feedback and the time you have invested in reviewing our work. We have carefully addressed all your comments in the rebuttal. If there are any remaining questions or points that could benefit from further clarification, we would be more than happy to provide additional details.
> >
> > Thanks again!
> >
> > Authors

---

### Official Review · Reviewer_Wt5f · 2025-10-28

**Soundness:** 3
**Presentation:** 2
**Contribution:** 3
**Rating:** 6
**Confidence:** 2

**Summary:**

This paper addresses the limitations of reinforcement learning from human feedback (RLHF) for preference alignment (PA) in low-resource environments. Standard RLHF requires many high-quality positive preference samples and can be unstable and computationally costly. The authors propose reframing PA as a machine unlearning (MU) problem: instead of learning from positives, the model can align with human preferences by selectively unlearning harmful negative samples.

They introduce U2A, a bi-level optimization framework that estimates the contribution of each negative sample to PA performance, then applies weighted unlearning to maximize alignment efficiency. Their theoretical analysis reveals two key insights. First, the effect of unlearning depends on whether the gradient direction of a sample conflicts with PA objectives. Negative samples with consistently low-reward behavior typically yield positive alignment gains when unlearned. Second, impact magnitude varies by sample and is a function of both inherent gradient norms and the chosen unlearning weight.

Experiments show that U2A significantly improves alignment under constrained resources while maintaining efficiency comparable to standard fine-tuning methods. The work offers a new perspective that connects PA and MU, opening new directions for optimizing alignment algorithms without heavy reliance on costly reward data.

**Strengths:**

The paper presents a clear line of reasoning from motivation to methodology and results. The insight that negative samples contribute unequally to preference alignment is both novel and practically valuable, especially in low-resource settings. The proposed U2A framework is grounded in bi-level optimization theory, enabling principled analysis of how individual negative samples should be weighted during unlearning. Experimental results demonstrate that selectively unlearning harmful samples leads to meaningful improvements in alignment efficiency and effectiveness, reinforcing the core contribution.

**Weaknesses:**

Some important elements of the method, including the primary algorithm and several key conceptual details, are placed in the appendix rather than the main paper. This makes the flow feel crowded in some areas while leaving gaps in others, and requires readers to frequently jump to supplementary materials to fully understand the approach. A more balanced layout would improve clarity and accessibility.

**Questions:**

How well does the proposed unlearning-based alignment hold up on larger, higher-quality models and more diverse preference datasets? A few comparison with the preference alignment using both positive and negative samples would be beneficial?

---

> ### Author Response · Authors · 2025-11-26
>
> >**Weakness:** Some important elements of the method are placed in the appendix rather than the main paper.
>
> **Answer:** Thank you for your valuable feedback. We apologize for any inconvenience caused by the layout and formatting, which may have affected your reading experience. Due to space constraints, we had to move some content to Appendix E. Once all revisions are completed, we will optimize the layout and move the content from Appendix E into the main text to improve the manuscript's readability.
>
>
> >**Question:** How well does the proposed unlearning-based alignment hold up on larger, higher-quality models and more diverse preference datasets? A few comparison with the preference alignment using both positive and negative samples would be beneficial?
>
> **Answer:** Thank you for your valuable feedback. For higher-quality models and more diverse preference datasets, we have conducted additional experiments using the Qwen 14B model as the backbone. As shown in Table 9 in Appendix F.5, the results demonstrate that U2A performs competitively on the Qwen 14B model, providing similar improvements in preference alignment as observed with the LLaMA models. This indicates that U2A's effectiveness is not limited to the LLaMA family and can extend to other powerful models as well.
>
> In addition, we observe that as the model size increases, the performance of U2A becomes more prominent. We believe this is because a larger model's scale pretraining confers stronger representational and generalization capabilities, enabling the model to infer positive behaviors and avoid harmful ones even when only negative samples are removed. In contrast, a smaller model's weaker generalization may lead to uncertainty or degraded behavior post-unlearning, impairing preference alignment performance. This aligns with the description we provided in lines 409-417 of the manuscript.
>
> For comparison with preference alignment using both positive and negative samples, it is worth noting that in preference alignment methods such as PPO and DPO, both positive and negative samples are used, with positive samples providing additional supervisory signals, thereby improving performance. However, in practical applications, the labeling cost of positive samples is relatively high. This is precisely the motivation for using unlearning on negative samples as an alternative to PPO and DPO methods for preference alignment. As shown in Table 10 of Appendix F.6, when only negative samples are used, our U2A framework achieves results comparable to those of PPO and DPO, which use both positive and negative samples. Furthermore, it can be observed that with the introduction of positive samples, the performance of U2A gradually improves. Under the condition of using the same positive samples as PPO and DPO, U2A significantly outperforms them. This suggests that using only negative samples for U2A preference alignment is a promising and cost-effective direction.
>
> We have included this new analysis in Appendix F.5 and F.6.

---

> > ### Author Response · Authors · 2025-11-28
> > **We welcome any remaining questions**
> >
> > Dear Reviewer Wt5f,
> >
> > Thank you for your valuable feedback and the time you have invested in reviewing our work. We have carefully addressed all your comments in the rebuttal. If there are any remaining questions or points that could benefit from further clarification, we would be more than happy to provide additional details.
> >
> > Thanks again!
> >
> > Authors

---

### Official Review · Reviewer_vu26 · 2025-10-30

**Soundness:** 3
**Presentation:** 2
**Contribution:** 2
**Rating:** 4
**Confidence:** 3

**Summary:**

The paper investigates how machine unlearning can be leveraged to improve preference alignment (PA) in large language models and proposes U2A (Unlearning to Align). It formulates unlearning-for-alignment as a bi-level problem: the inner loop performs unlearning on selected training samples with a chosen weight, while the outer loop measures the resulting change in PA quality. Using implicit differentiation, the paper shows that the PA change from unlearning a sample depends on gradient alignment between the PA objective and the unlearning loss; unlearning can help or harm PA depending on this alignment and the chosen weight, motivating selective, weighted unlearning rather than blanket removal.

U2A then picks a small set of impactful samples and assigns weights, using a greedy selector plus simple weight updates; it exploits efficient Hessian-vector products to keep the cost manageable. On SafeRLHF, UltraFeedback, and HaluEval with LLaMA-2/3, U2A improves preference alignment and unlearning metrics over prior heuristics (GA, GradDiff, NPO) and is competitive with PPO/DPO baselines.

**Strengths:**

Originality: The paper reframes machine unlearning as a means to improve preference alignment rather than only a privacy safeguard. Casting “unlearning to align” as a bi-level problem with an explicit criterion for whether removing a sample helps or hurts alignment is a fresh, principled alternative to heuristic data filtering.

Quality: The technical pathway from the bi-level formulation to a practical algorithm is coherent. The use of a first-order influence-style approximation to estimate alignment impact, plus a selective procedure (greedy/matching-pursuit) with simple weight updates (mirror descent) and efficient Hessian–vector products, makes the method implementable at scale and links theory to practice.

Clarity: The paper communicates the core intuition clearly: unlearning helps when the unlearning and alignment gradients point in similar directions and can harm otherwise. The pipeline—outer evaluation of alignment, inner weighted unlearning, then selection updates is laid out in a way that readers can follow and reproduce.

Significance: Results across multiple datasets and backbones indicate consistent gains over common unlearning heuristics and competitive performance with strong alignment baselines. The approach offers a data-centric knob that can complement or reduce reliance on heavier RLHF-style training, suggesting practical value for teams maintaining alignment under evolving safety and content policies.

**Weaknesses:**

The paper’s structure is difficult to follow. Sections frequently interleave intuition, which obscures the main thread of the method. The evaluation is also hard to parse: the metrics used in tables and figures are not defined where they first appear, and captions do not restate what higher/lower means or why a difference is meaningful. Baselines are listed with names only, without a brief description or implementation details, so readers cannot judge comparability.

The reported win rates show little or no strengthening over baselines (especially Table 2). Given the small margins, the evidence is insufficient to claim a meaningful improvement. As LLM-as-judge metrics can be noisy and prompt-sensitive, these results should be complemented with stronger analysis.

**Questions:**

1. Assumption 3.1 (mildness and applicability). Could the authors precisely restate Assumption 3.1 and explain whether it is a mild assumption in practice? Please discuss concrete conditions under which modern LLMs (and the proposed training setup) satisfy it. Clarify how the algorithm remains stable when the assumption is violated.

2. Model family dependence (beyond Llama): All experiments use Llama backbones. Please either include an additional LLM (e.g., Mistral, Qwen, Gemma) or provide an analysis explaining why the method is expected to transfer.

3. Computational overhead of selection. What is the additional cost introduced by the dynamic selection and weighting procedure?
Please report GPU-hours, compared to baselines.

---

> ### Author Response · Authors · 2025-11-26
>
> >**Weakness 1:** The paper’s structure is difficult to follow.
>
> **Answer:** Thank you for your valuable feedback. We apologize for any inconvenience caused by the layout and formatting, which may have affected your reading experience. Due to space constraints, we had to move some content to Appendix E. Once all revisions are completed, we will optimize the layout and move the content from Appendix E into the main text to improve the manuscript's readability.
>
> >**Weakness 2:** The reported win rates show little or no strengthening over baselines (especially Table 2).
>
> **Answer:** The unlearning method based on the U2A improvement can achieve comparable results to traditional PA methods, such as PPO and DPO. The reason it does not significantly outperform PPO and DPO is that the latter methods use additional positive examples as guiding signals, but the high cost of positive examples led to their exclusion in our setup.
>
> To further validate U2A's practicality, we conduct an additional experiment that introduces positive examples into the U2A framework. More specifically, we incorporate various proportions of positive examples (k%) during the U2A training process. The comparison results with PPO and DPO, as shown in Table 10 in Appendix F.6, clearly demonstrate that the performance of U2A improves continuously as the number of positive examples increases. Moreover, under the same number of positive examples, U2A consistently outperforms both PPO and DPO, indicating that introducing positive sample guidance significantly enhances preference alignment.
>
> Additionally, as model size increases, the performance of U2A also strengthens, as shown in Tables 9 and 10. We believe this is because a larger model's scale pretraining confers stronger representational and generalization capabilities, enabling the model to infer positive behaviors and avoid harmful ones even when only negative samples are removed. In contrast, a smaller model's weaker generalization may lead to uncertainty or degraded behaviors after unlearning, impairing preference alignment performance. This aligns with the description we provided in lines 409–417 of the manuscript, suggesting that using only negative samples for U2A to perform preference alignment is a promising and cost-effective direction.
>
> We have included this new analysis in Appendix F.6.
>
> >**Question 1:** Assumption 3.1 (mildness and applicability).
>
> **Answer:** We have revised Assumption 3.1 in the manuscript to provide a clearer definition. From a theoretical perspective, Assumption 3.1 aligns with classical optimization assumptions, and the conditions on the differentiability of the loss functions $L$ and $\mathcal{J}$ are standard in machine learning literature [1]. Moreover, the assumption of the semi-positive definiteness of the Hessian matrix of $L$ is commonly applied in research on machine unlearning [2,3]. This assumption is mild and fits within the unlearning framework for LLMs.
>
> We further clarify this assumption in three dimensions:
>
> 1. Modern LLMs generally follow a pretraining-then-finetuning strategy. During the finetuning phase, model parameters have typically converged to a favorable local region, with a smooth loss surface and controllable curvature. For example, [1] pointed out that finetuned models exhibit controllability in terms of Hessian matrix eigenvalues and gradient norms, enabling the derivation of generalization bounds. In this state, assuming that the loss function corresponding to each sample behaves well (differentiable and the Hessian matrix is close to semi-positive definite) around the model parameters is a reasonable approximation.
>
> 2. In machine unlearning or preference alignment scenarios, removing samples generally involves operations like "a few negative samples" and "small-intensity perturbations," rather than reconstructing large-scale training data. Such operations imply that model parameters undergo only local adjustments, making local analysis assumptions (e.g., the semi-positive definiteness of the Hessian matrix) more likely to hold. For instance, recent research [2] has indicated that one of the prerequisites for "approximate unlearning" methods based on residual-influence functions or Hessian matrix estimation is that the Hessian matrix is estimable, and the model remains stable locally.

---

> ### Author Response · Authors · 2025-11-26
>
> 3. In many alignment or unlearning scenarios, negative samples typically represent "responses generated by the model that deviate significantly from human preferences" or "sensitive/undesirable data." These samples can be modeled in the loss function with simple but highly penalizing terms. In such cases, the behavior of the loss function with respect to the model parameters is more likely to exhibit convexity and smoothness, with the Hessian matrix close to semi-positive definite. Therefore, from a practical standpoint, assuming that the loss function satisfies these conditions does not imply a "strong assumption" but rather an assumption that "naturally holds in many well-designed deletion strategies."
>
> While extreme cases, such as strong curvature, may violate this assumption, the algorithm can still maintain stability through various mechanisms: using small learning rates or small-intensity perturbations $\omega$ to keep adjustments within a near-convex region; regularization to limit the distance between updated and original parameters, reducing the risk of high-curvature regions; phased or batch unlearning strategies, where a small number of samples are initially removed, with monitoring to detect performance degradation or oscillations, as validated in [4].
>
> >**Question 2:** Model family dependence (beyond Llama).
>
> **Answer:** Thank you for your suggestion. To evaluate the generalizability of our method across different model families, we have included an additional large language model, namely the Qwen 14B model, as the backbone in our experiments. As shown in Table 9 in Appendix F.5, the results demonstrate that U2A performs competitively on the Qwen 14B model, providing similar improvements in preference alignment as observed with the LLaMA models. This indicates that U2A's effectiveness is not limited to the LLaMA family and can extend to other powerful models as well. In addition, we observe that as the model size increases, the performance of U2A becomes more prominent, as we addressed in Weakness 2.
>
> We have included this new result in Appendix F.5 for further clarification.
>
>
> >**Question3:** Computational overhead of selection.
>
> **Answer:** Thank you for your valuable question. To evaluate the computational efficiency of our proposed U2A framework, we conduct a time efficiency assessment, which is detailed in Appendix F.4. In this evaluation, we compare the training time and computational overhead of U2A with standard PPO and DPO baselines on two datasets (SafeRLHF and UltraFeedback) and two model sizes (LLaMA2 and LLaMA3). The results, shown in Table 8, demonstrate that U2A performs competitively in terms of both training time and sample efficiency.
>
>
> #### **Reference**
>
> [1] Ju, Haotian, Dongyue Li, and Hongyang R. Zhang. "Robust fine-tuning of deep neural networks with hessian-based generalization guarantees." ICML 2022.
>
> [2] Nguyen, Thanh Tam, et al. "A survey of machine unlearning." ACM Transactions on Intelligent Systems and Technology 2025.
>
> [3] Ahmed, Sk Miraj, et al. "Towards Source-Free Machine Unlearning." CVPR 2025.
>
> [4] Oesterling, Alex, et al. "Fair machine unlearning: Data removal while mitigating disparities." International Conference on Artificial Intelligence and Statistics 2024.

---

> ### Author Response · Authors · 2025-11-28
> **Please let us know if anything remains unclear**
>
> Dear Reviewer vu26,
>
> Thank you for your thoughtful review and constructive suggestions. We have thoroughly addressed all the questions raised in your review. Should any part of our response appear unclear or incomplete, please let us know! We would be more than willing to clarify or provide additional details.
>
> We sincerely appreciate your effort during the review process.
>
> Authors

---

### Official Review · Reviewer_AXVa · 2025-11-01

**Soundness:** 2
**Presentation:** 2
**Contribution:** 2
**Rating:** 4
**Confidence:** 3

**Summary:**

The paper proposes to connect machine unlearning (MU) with preference alignment (PA) for LLMs via a bi-level formulation that measures how unlearning specific negative samples changes a PA objective. From this analysis, the authors design Unlearning to Align (U2A), which selects and weights negative examples to unlearn, aiming to maximize PA. Experiments on SafeRLHF and UltraFeedback with Llama-2/3 variants show that U2A improves over full-set unlearning baselines and is competitive with DPO/PPO in some settings.

**Strengths:**

- A new viewpoint of PA into unlearning seems interesting
- Consistent improvements of GA/GradDiff/NPO with U2A plugged in supports the importance of sample-wise weighting of the negative samples.

**Weaknesses:**

- Proposition 3.1 involves $\nabla L_{\text{forget}}(x)$, but there are no concrete properties of the training sample $x$. Conclusion 1-2 become trivial restatements abour cosine similarity and gradient norm. The proposition should derive sample-dependent bounds under further properties of training sample $x$.
- Justification on "negative-only is cheaper": There are no evidence on actual costs, noise, or coverag for user reports/red-team data. As this work's motivation stems from this problem. It should be further justified.
- Although U2A consistently improves the unlearning baseine, it's efficacy hinders when compared with basic PA methods. This result does not seem to strongly support U2A's practicality.

**Questions:**

- Could you provide a intuitive elaboration as to why U2A helps unlearning methods beneficial to PA?

---

> ### Author Response · Authors · 2025-11-26
>
> >**Weakness 1:** Proposition 3.1 involves $\nabla L_{\text{forget}}(x)$, but there are no concrete properties of the training sample $x$. Conclusion 1-2 become trivial restatements about cosine similarity and gradient norm. The proposition should derive sample-dependent bounds under further properties of the training sample $x$.
>
> **Answer:** Thank you for your valuable suggestions. We have revised the relevant sections of the manuscript based on your feedback, changing the term "conclusion" to "fact," as our original intention was merely to present this observation as the motivation for proposing U2A, rather than attempting to quantify the relationship between sample characteristics and PA performance. Given the non-linearity and non-convexity of LLMs and the complexity of data distributions, it is unrealistic to quantify this relationship theoretically without strong assumptions.
>
> Additionally, we have included further formal analysis in the manuscript. In Theorem C.2 of Appendix C.2, we theoretically prove that if there is room for improvement in the PA objective, a reweighting scheme $\boldsymbol{\omega}$ exists that can improve the PA objective without increasing the MU task loss, thus preserving multitask performance. This property provides the theoretical foundation for our subsequent algorithm design.
>
> >**Weakness 2:** Justification on “negative-only is cheaper”: There are no evidence on actual costs, noise, or coverage for user reports/red-team data. As this work's motivation stems from this problem. It should be further justified.
>
> **Answer:** The reasoning behind “only negative samples are cheaper” stems from the comparison with “high labeling costs for positive samples.” Existing preference alignment methods typically rely on positive-negative sample pairs, but the labeling process for positive samples often requires human intervention, making it costly [1]. Therefore, using only negative samples is obviously more cost-effective compared to using both positive and negative samples.
>
> Additionally, recent research [2,3] has shown that obtaining negative samples is generally less difficult than obtaining positive samples. As a result, compared to traditional positive-negative sample pairs, using only negative samples offers significant advantages in terms of practicality and cost-effectiveness.
>
> >**Weakness 3:** Although U2A consistently improves the unlearning baseine, its efficacy hinders when compared with basic PA methods. This result does not seem to strongly support U2A's practicality.
>
> **Answer:** The unlearning method based on the U2A improvement can achieve comparable results to traditional PA methods, such as PPO and DPO. The reason it does not significantly outperform PPO and DPO is that the latter methods use additional positive examples as guiding signals, but the high cost of positive examples led to their exclusion in our setup.
>
> To further validate U2A's practicality, we conduct an additional experiment that introduces positive examples into the U2A framework. More specifically, we incorporate various proportions of positive examples (k%) during the U2A training process. The comparison results with PPO and DPO, as shown in Table 10 in Appendix F.6, clearly demonstrate that the performance of U2A improves continuously as the number of positive examples increases. Moreover, under the same number of positive examples, U2A consistently outperforms both PPO and DPO, indicating that introducing positive sample guidance significantly enhances preference alignment.
>
> Additionally, as model size increases, the performance of U2A also strengthens, as shown in Tables 9 and 10. We believe this is because a larger model's scale pretraining confers stronger representational and generalization capabilities, enabling the model to infer positive behaviors and avoid harmful ones even when only negative samples are removed. In contrast, a smaller model's weaker generalization may lead to uncertainty or degraded behaviors after unlearning, impairing preference alignment performance. This aligns with the description we provided in lines 409–417 of the manuscript, suggesting that using only negative samples for U2A to perform preference alignment is a promising and cost-effective direction.
>
> We have included this new analysis in Appendix F.6.

---

> ### Author Response · Authors · 2025-11-26
>
> >**Question:** Could you provide a intuitive elaboration as to why U2A helps unlearning methods beneficial to PA?
>
> **Answer:** As shown in Figures 1 and 2, given a set of samples to be forgotten, which includes both low-reward and high-reward samples, unlearning low-reward samples can be seen as pushing the model in the opposite direction of low rewards, thereby improving PA performance. On the other hand, unlearning high-reward samples pushes the model in the opposite direction of high rewards, leading to a decrease in PA performance. The role of U2A is to identify the samples that most significantly improve PA performance, and based on the impact of unlearning these samples on PA performance, it assigns a corresponding unlearning weight to each sample, thus maximizing the improvement in PA performance.
>
>
> #### **Reference**
>
> [1] Jiang, Ruili, et al. “A survey on human preference learning for large language models.” arXiv 2024.
>
> [2] Yao, Yuanshun, Xiaojun Xu, and Yang Liu. “Large language model unlearning.” NeurIPS 2024.
>
> [3] Duan, Shitong, et al. “Negating Negatives: Alignment with Human Negative Samples via Distributional Dispreference Optimization.” EMNLP 2024.

---

> ### Author Response · Authors · 2025-11-28
> **Please feel free to request any further clarification**
>
> Dear Reviewer AXVa,
>
> Thank you very much for the time you have devoted to reviewing our paper. We have carefully addressed all the concerns you raised and provided detailed explanations in the rebuttal. If any part of our response remains unclear or requires further elaboration, please tell us, we would be grateful to clarify it.
>
> Thanks again!
>
> Authors

---

> > ### Comment · Reviewer_AXVa · 2025-11-28
> >
> > Thank you for the authors’ detailed response and added analysis. However, I remain unconvinced that the claim “negative-only sampling is clearly cheaper/better than pairwise (positive-negative) data” is sufficiently supported. In many prior works on preference alignment, the use of pairwise preference labels reduces annotation burden and cost compared to rating- or score-based datasets. Thus, if you argue for using only negative/absolute labels instead of pairwise data, you should provide stronger, empirical justification for why that less burdensome preference-labeling approach should be discarded.

---

> > > ### Author Response · Authors · 2025-11-28
> > >
> > > Thank you for your response. Perhaps there was some lack of clarity in our expression, which led to a different interpretation on your part. The point we made that "using only negative samples is clearly cheaper than using paired (positive-negative) data" is, in our view, self-evident.
> > >
> > > For example, suppose our goal is to achieve preference alignment. Using 1000 negative samples would be sufficient to achieve this goal; whereas using 1000 positive-negative sample pairs (i.e., 1000 positive samples and 1000 negative samples, totaling 2000 samples) would also achieve the same outcome. **Clearly, the cost of 1000 negative samples is lower than that of 1000 positive-negative sample pairs, because the latter includes an additional 1000 positive samples, and the labeling cost for positive samples is higher.** This point is widely recognized in the field of preference alignment.
> > >
> > > Additionally, your comment about "advocating the use of only negative/absolute labels" seems to suggest that negative samples also need to be labeled. **However, in reality, negative samples do not require labeling, as they can be generated through red-teaming automated tests.** This has been widely discussed in existing research on LLM red-team attacks. In contrast, positive samples require manual labeling (e.g., scoring or preference tagging), which incurs a higher cost.
> > >
> > > Our experiments in the manuscript also confirm this. Under the condition of controlling the sample size, the U2A method using N negative samples achieves similar results to the PA method using N positive-negative sample pairs.
> > >
> > > We hope this explanation effectively addresses your concerns, and we look forward to further discussion.

---

### Author Response · Authors · 2025-11-27
**Global response**

We would like to thank all reviewers for their valuable feedback and for acknowledging the novelty of "Machine Unlearning from the Perspective of Preference Alignment!" We revised our initial submission in response to their comments as follows:

- Formal introduction of the theoretical analysis of U2A in **Appendix C.2** in response to **Reviewer AXVa**.
- Additional experimental analysis of U2A on the larger non-Llama model Qwen 2.5 14B in **Appendix F.5** in response to **Reviewers vu26 and Wt5f**.
- Additional experimental analysis comparing the PA performance of U2A under the positive sample condition with PPO and DPO in **Appendix F.6** in response to **Reviewers AXVa, Wt5f, and MF9N**.
- Modifications to two statements in response to **Reviewer AXVa and vu26**.

---

### Author Response · Authors · 2025-12-02
**Concluding response**

We would like to sincerely thank all the reviewers for their valuable feedback and positive evaluations. Below, we summarize the key points of the discussion.

All reviewers acknowledged **the novelty of approaching the issue of machine unlearning from the perspective of preference alignment**. In response to the reviewers' comments, we have made the following updates: (1) We have provided a more detailed theoretical analysis to clarify the motivation for introducing the U2A method; (2) We have conducted further empirical validation on larger, more diverse models (such as Qwen 2.5 14B) and datasets; (3) We have performed quantitative experiments comparing the performance of the U2A method with additional positive samples, demonstrating its potential for application. Our additional experiments further support and strengthen the main findings.

It is worth noting that reviewer AXVa seems to have misunderstood our work, believing that the claim "negative sampling is significantly more cost-effective than paired (positive-negative) data" was insufficiently justified. However, we believe this conclusion is self-evident, as clarified in our Official Comment below. This point is also one of the paper's core motivations: by unlearning from negative samples, we can achieve preference alignment across both positive and negative samples. Given the high cost of labeling positive samples, machine unlearning significantly reduces the overhead in this context.

Overall, we wish to emphasise that the main contribution of this work is the **first integration of two previously independent research areas—machine unlearning and preference alignment—resulting in a new and effective machine unlearning framework**. We believe this novel perspective offers significant value to both research communities and opens up promising directions for future research.

---

### Note · Authors · 2026-01-26

I have read and agree with the venue's withdrawal policy on behalf of myself and my co-authors.

---

### Meta-Review · Area_Chair_gNGy · 2026-01-11

**Summary:**

Summary of reviewers' concerns:

- Proposition 3.1 involves $\nabla L\_{\mathrm{forget}}(x)$, but there are no concrete properties of the training sample $x$. Conclusion 1-2 become trivial restatement. The proposition should derive sample-dependent bounds.

- The motivation on "negative-only is cheaper" is still not very clear.

- U2A's efficacy hinders when compared with basic PA methods. This result does not seem to strongly support U2A's practicality. The reported win rates show little or no strengthening over baselines (especially Table 2). While U2A is competitive in some settings, it trails DPO/PPO in others (e.g., with weaker base models / noisier negatives). A controlled compute-matched study would clarify trade-offs.

- The paper's structure is difficult to follow. Sections frequently interleave intuition, which obscures the main thread of the method. Some important elements are placed in the appendix rather than the main paper.  The evaluation is also hard to parse. Baselines are listed with names only, without a brief description or implementation details.

- As LLM-as-judge metrics can be noisy and prompt-sensitive, these results should be complemented with stronger analysis.

- Explain whether Assumption 3.1 is a mild assumption in practice. Discuss concrete conditions under which modern LLMs satisfy it. Clarify how the algorithm remains stable when Assumption 3.1 is violated.

- All experiments use Llama backbones. Please consider an additional LLM (e.g., Mistral, Qwen, Gemma).

- What is the additional cost introduced by the dynamic selection and weighting procedure? Please report GPU-hours, compared to baselines.

- How well does the proposed unlearning-based alignment work on larger, higher-quality models and more diverse preference datasets, such as safety red-teaming, multilingual, and domain tasks? Comparisons with the preference alignment using both positive and negative samples would be beneficial.

- The theory uses Hessian PSD & diagonal approximations; stress-testing against curvature misspecification would be valuable.

**Reviewer Concerns:**

We appreciate that the authors have addressed many of the reviewers' concerns by adding more experiments, revising some technical content for better clarity, and providing additional explanations or theoretical analysis. However, the current rebuttal still does not fully address some of the reviewers' concerns, mainly in the following aspects: (1) Reviewers vu26 and Wt5f both pointed out that the paper is difficult to follow. For example, sections frequently interleave intuition, which obscures the main thread of the method. Some important elements are placed in the appendix. The evaluation is also hard to parse. Baselines are listed with names only, without a brief description or implementation details. (2) Reviewer AXVa remains unconvinced that the claim "negative-only sampling is clearly cheaper/better than pairwise (positive-negative) data" is sufficiently supported. The authors mentioned red-teaming automated tests for negative data. It would be interesting to discuss whether pairwise data could also be automatically annotated using a well-designed prompt and LLMs. The authors can provide more concrete empirical evidence or a realistic cost comparison of negative-only vs. pairwise labeling to justify this paper's motivation. (3) Reviewer MF9N requested evaluation on more diverse datasets (e.g., safety red‑teaming, multilingual, and domain‑specific tasks). In the rebuttal, the authors state that relevant methods might not be easily applicable to these settings. This justification is somewhat not fully convincing. The paper would be strengthened by including at least some additional experiments, even on datasets different from those explicitly mentioned by the reviewer, to provide evidence for the method’s generality. If such experiments are truly infeasible, the authors should describe the specific constraints in more detail, rather than relying on a relatively unclear statement such as "not be easily applicable." Consequently, this response does not fully resolve the concern about the generalization.

**Reviewer Scores:**

Reviewers Wt5f and MF9N gave initial scores of 6. Reviewers AXVa and vu26 gave initial scores of 4. As discussed above, the authors do not fully address Reviewers AXVa, vu26, and Wt5f's concerns. Thus, I think the three reviewers are unlikely to change their ratings, leaving this paper below the borderline.

---

### Decision · Program_Chairs · 2026-01-26

Reject